# Language models enable zero-shot prediction of the effects of mutations on protein function

**Joshua Meier** [1] [2]   **Roshan Rao** [3]   **Robert Verkuil** [1]   **Jason Liu** [1]
**Tom Sercu** [1]   **Alexander Rives** [1] [2]

## Abstract

Modeling the effect of sequence variation on function is a fundamental problem for understanding and designing proteins. Since evolution encodes information about function into patterns in protein sequences, unsupervised models of variant effects can be learned from sequence data. The approach to date has been to fit a model to a family of related sequences. The conventional setting is limited, since a new model must be trained for each prediction task. We show that using only zero-shot inference, without any supervision from experimental data or additional training, protein language models capture the functional effects of sequence variation, performing at state-of-the-art.

## 1 Introduction

Proteins have a myriad of diverse functions that underlie the complexity of life. Protein sequences encode function via structure through the spontaneous folding of the sequence into the three dimensional structure of the protein [1]. The effects of sequence mutations on function form a landscape that reveals how function constrains sequence. Alterations at some sites in a protein sequence cannot be tolerated because they are essential to the protein's function. Other sites evolve together because the structure and function is determined by them collectively. Mutations can enhance the activity of a protein, attenuate it, or leave it unchanged.

The functional effect of sequence variations can be measured through deep mutational scanning experiments [2]. Consisting of thousands to hundreds of thousands of measurements of protein function, deep mutational scans give insight into the intrinsic constraints on a protein's structure and function. Due to the cost and difficulty of implementing such experiments, compilations of deep mutational scanning data include experiments on a few dozens of proteins at most, relative to the tens of thousands of proteins encoded in the human genome, and the millions more across the tree of life that we would like to understand.

A model that learns the landscape linking sequence to function can provide insight into function without having to do experiments. Unsupervised models of mutational effects can be learned from sequences [3, 4]. Statistical patterns in a family of evolutionarily related protein sequences contain information about structure and function [5–7]. This is because the properties of a protein act as constraints on the selection of sequences through evolution [8].

In the natural language modeling community, there has been interest in zero-shot transfer of models to new tasks. Massive language models can solve tasks they haven't been directly trained on [9–11]. Recently protein language models have achieved state-of-the-art in various structure prediction tasks [12–14]. Work to date has mainly focused on transfer in the classical representation learning setting, using pre-trained features with supervision on the downstream task.

---

[1] Facebook AI Research [2] New York University [3] UC Berkeley. ESM-1v is available at <https://github.com/facebookresearch/esm>. Correspondence to: Alexander Rives <arives@fb.com>.

35th Conference on Neural Information Processing Systems (NeurIPS 2021).

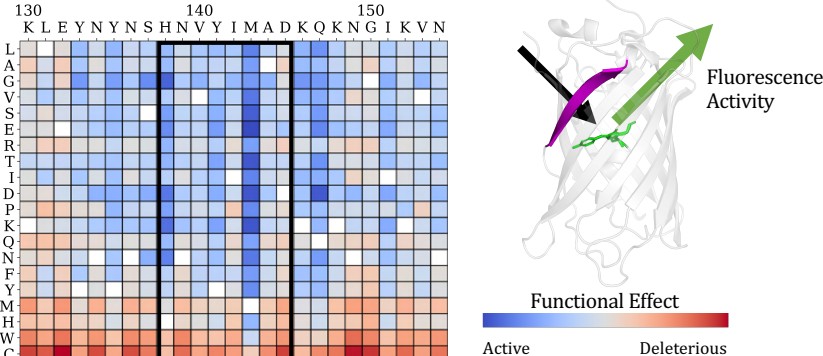

Figure 1: Depiction of a mutational effect prediction task. The objective is to score the effect of sequence mutations on the function of a protein. Deep mutational scanning experiments provide ground truth experimental measurements of the protein's function (fluorescence activity in the example here) for a large set of single mutations or combinations of mutations. For each protein, the prediction task is to score each possible mutation and rank its relative activity. Predictions for single substitutions can be described in a score matrix. The columns are the positions in the sequence. The rows are the possible variations at each position.

In this work we show that language models trained on large and diverse protein sequence databases can predict experimental measurements of protein function without further supervision. Prior work has focused on transferring the representations using supervision from experimental data [15, 16]. We find that language models can transfer to predict functional measurements without supervision. Language models perform *zero-shot* and *few-shot* prediction of mutational effects across a variety of proteins with widely differing functions. We perform experiments with state-of-the-art protein language models ESM-1b [12] and MSA Transformer [13]. We introduce a new protein language model, ESM-1v, with zero-shot performance comparable to state-of-the-art mutational effect predictors. Performance can be further improved by fine-tuning the model with sequences from the protein family. Predictions capture the functional landscape of the protein, correlate with amino acid conservation patterns in the core and surface, and identify residues responsible for binding and activity.

## 2 Zero-shot transfer

Zero-shot learning has classically described the extension of a classifier to a new set of classes that have not been seen in training [17]. In natural language processing this idea has been extended to describe the transfer of models to entirely new tasks without further training. Proposed as zero-data learning by Larochelle et al. [18], this perspective on transfer has been at the center of recent work understanding the generalization capabilities of large language models [9–11, 19]. The distinction from representation learning is that the models are used *directly* without additional supervision for the task. This means that the tasks must be learned purely from pre-training.

In this work we take a similar perspective on zero-shot transfer to that of GPT-3, described in Brown et al. [10]. We define zero-shot transfer to be transfer of a model to a new task without any further supervision to specialize the model to the task. We also consider the closely related idea of few-shot transfer. Here as in Brown et al. [10] we define the few-shot setting to be one in which a few positive examples are given to the model as inputs at inference time. As in the zero-shot setting, no gradient updates are performed to specialize the model. Similar to Brown et al. [10], the claim is not one of out-of-distribution generalization. The assumption is that in the pre-training stage, the model learns information relevant to the tasks to which it will later be transferred. In the case of protein language models, the pre-training dataset includes sequences from across evolution, which implies the model may see examples of sequences from protein families on which it will be evaluated. The essential

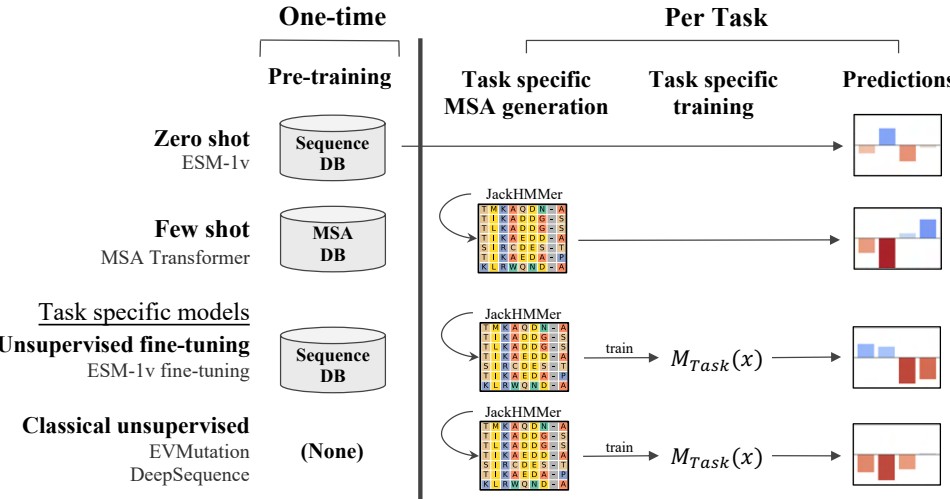

Figure 2: Steps involved in variant effect prediction methods. Compared with EVMutation [4] and DeepSequence [20], MSA Transformer and ESM-1v require no task-specific model training for inference. Moreover, ESM-1v does not require MSA generation.

departure from the standard approach in computational biology is that the model is *general purpose* and can be applied across a variety of tasks without specialization.

Measurements of function, a property of central importance to the understanding and design of proteins, are a practical ground for studying the generalization capability of protein language models. Deep mutational scanning experiments measure the effects of thousands to hundreds of thousands of mutations on a single protein, and have been performed on a variety of proteins having different functions and using various forms of experimental measurement. We study zero-shot and few-shot transfer of protein language models to function prediction using this data.

Supervised methods trained with data from experimental measurements [15, 16], and unsupervised methods trained only on sequences [3, 4] have been developed for prediction of mutational effects. Unsupervised mutational effect predictors are trained as task specific models on sequences from an individual protein family. In this view every protein is an independent prediction task where the objective is to score the effect of mutations on the protein's function. While mutational effect predictors trained on multiple sequence alignments (MSAs) are typically described as unsupervised, they can also be seen as weakly supervised. Hsu et al. [15] observe that such models have weak supervision on the task through the MSA, which describes the fitness landscape of the protein through positive examples.

If protein language models can learn the information necessary to solve a task from pre-training, then they can be applied *directly* to new instances of the task, without specialization. This would mean that in practice a single general purpose model can be trained once and then applied to a variety of possible tasks. Thus zero-shot and few-shot transfer represent fundamentally new unsupervised learning capabilities that protein language models can bring to the computational biology toolkit.

# 3 Method

Protein language models trained with the masked language modeling objective are supervised to output the probability that an amino acid occurs at a position in a protein given the surrounding context. We use this capability to score sequence variations. For a given mutation we can consider the amino acid in the wildtype protein as a reference state, comparing the probability assigned to the mutated amino acid with the probability assigned to the wildtype.

| Models | Full | Test |
|---|---|---|
| PSSM | 0.460 | 0.460 |
| EVMutation (published) | 0.508 | 0.495 |
| EVMutation (replicated) | 0.511 | 0.498 |
| DeepSequence (published) | 0.514 | 0.499 |
| DeepSequence (replicated) | 0.520 | 0.506 |
| MSA Transformer | 0.542 | 0.524 |
| ESM-1v (zero shot) | 0.509 | 0.482 |
| ESM-1v (+further training) | 0.538 | 0.519 |

Table 1: Comparison of protein language models to state-of-the-art methods. Average |Spearman $\rho$| on full and test sets. DeepSequence and ESM-1v models are each ensembles of 5 models. MSA Transformer is a single model, but is ensembled across 5 random samples of the MSA.

We score mutations using the log odds ratio at the mutated position, assuming an additive model when multiple mutations $T$ exist in the same sequence:

$$\sum_{t \in T} \log p(x_t = x_t^{mt}|x_{\setminus T}) - \log p(x_t = x_t^{wt}|x_{\setminus T}) \tag{1}$$

Here the sum is over the mutated positions, and the sequence input to the model is masked at every mutated position.

### 3.1 Zero-shot and few-shot transfer

In the zero-shot setting, inference is performed directly on the sequence to be evaluated. Since the MSA Transformer can take multiple sequences as input at inference time, we use this model in the few-shot setting, where additional sequences from the protein family are provided along with the sequence to be evaluated. In both the zero-shot and few-shot settings, only forward passes of the models are performed during inference; no gradient updates are taken. Fig. 2 illustrates the approach in comparison to the current practice of fitting a new model for each task.

### 3.2 Inference efficiency

Inference with ESM-1v is more efficient than current state-of-the-art methods. This is a result of two important differences: (i) the effect of mutations can be inferred directly without training a task-specific model; (ii) fitness landscapes can be predicted with a single forward pass. Time requirements are summarized in Fig. 7.

### 3.3 Scoring with MSA Transformer

We score mutations with MSA Transformer using the log odds ratio and additive model in Eq. (1). However, since MSA Transformer uses a set of sequences for inference, we input the sequence to be evaluated as the first sequence, and provide additional sequences from the MSA as context. Masking and scoring are performed on the first sequence only.

## 4 Results

### 4.1 Experimental setup

**Prediction Models**    We compare to state-of-the-art unsupervised variant prediction methods, EV-Mutation [4] and DeepSequence [20]. We also examine performance of a variety of protein language models that have been recently introduced in the literature.

The position specific scoring matrix (PSSM), EVmutation [4], and DeepSequence [20] methods are all MSA based. The PSSM treats each position in the sequence independently, factorizing the likelihood into one term per sequence position. EVmutation is a Potts model, which adds pairwise terms modeling the interactions between positions. DeepSequence introduces a latent code, allowing potential higher-order interactions between positions.

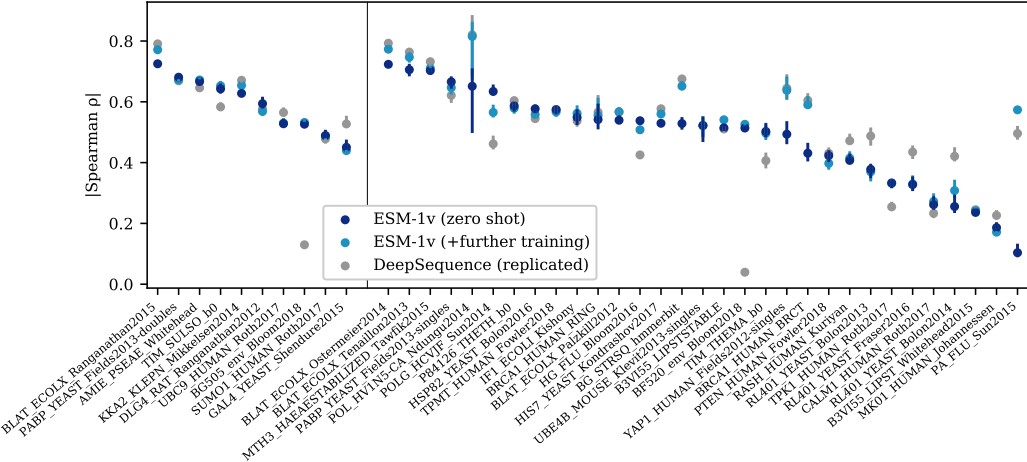

Figure 3: Per task performance. Comparison across 41 deep mutational scanning datasets. Points are |Spearman $\rho$| on each dataset, error bars show standard deviation of 20 bootstrapped samples. Validation proteins are shown to the left of the dividing line and test proteins to the right. In 17 out of the 41 tasks, ESM-1v zero-shot has a higher |Spearman $\rho$| than DeepSequence.

UniRep [21], TAPE [22], ProtBERT-BFD [14], ESM-1b [12], and ESM-1v (introduced here), are all single-sequence language models trained on large databases of unaligned and unrelated protein sequences (e.g. Pfam [23] or UniRef [24]). With the exception of UniRep, which is trained using next token prediction, all models are trained with masked language modeling [25].

Finally, the MSA Transformer [13] is a combination of both approaches; it is trained on a large database of MSAs using masked language modeling and takes an MSA as input during inference.

**ESM-1v**   We train ESM-1v, a 650M parameter transformer language model for prediction of variant effects, on 98 million diverse protein sequences across evolution. The model is trained only on sequences, without any supervision from experimental measurements of function. We use Uniref90 2020–03 [24], employing the ESM-1b architecture and masked language modeling approach of Rives et al. [12]. The model attains a perplexity of 7.29 on a set of held-out Uniref90 sequences (Table 10). We train five models with different seeds to produce an ensemble.

**Evaluation**   Models are evaluated on a set of 41 deep mutational scans collected by Riesselman et al. [20], which comprise a variety of tasks assessing a diverse set of proteins. Across tasks, the experiments differ in the functions tested and in the measurements performed. We treat each deep mutational scanning dataset as a separate prediction task, scoring each of the variants in the dataset with the model. The tasks are split into a validation set of ten mutational scanning datasets and a test set consisting of the remaining datasets. We evaluate performance by comparing the scores with the experimental measurements using Spearman rank correlation.

**Comparisons**   Since the published versions of EVMutation and DeepSequence use MSAs generated from an earlier version of Uniref100, we generate new MSAs using EVMutation methodology and the version of Uniref100 concurrent with our pretraining dataset. We train replications of EVMutation and DeepSequence using their open source code. The same MSAs are also used in few-shot experiments with MSA Transformer and unsupervised fine-tuning experiments with ESM-1v.

### 4.2   Language models enable zero-shot and few-shot prediction of the effects of mutations

ESM-1v and MSA Transformer models make state-of-the-art predictions. Table 1 compares overall performance of the models across the 41 mutational scanning datasets. Fig. 3 presents a comparison between ESM-1v and DeepSequence on each of the tasks. Zero-shot inference with ESM-1v has a better correlation with experimental measurements than DeepSequence on 17 of the 41 datasets. The two methods are not statistically distinguishable via a paired *t*-test.

| Models | Full | Test |
|--------|------|------|
| UniRep | 0.156 | 0.151 |
| TAPE | 0.171 | 0.175 |
| ProtBERT-BFD | 0.428 | 0.399 |
| ESM-1b | 0.459 | 0.424 |
| ESM-1v[†] | 0.484 | 0.457 |
| ESM-1v[*] | 0.509 | 0.482 |

Table 2: Zero-shot performance. Average |Spearman $\rho$| on full and test sets. [†]Average performance of five ESM-1v models. [*]Ensemble of the five ESM-1v models.

Table 2 compares protein language models in the zero-shot setting. ESM-1v outperforms existing protein language models TAPE [22], UniRep [21], ProtBERT-BFD [14], and ESM-1b [12]. Fig. 8 breaks down performance across each of the tasks.

**Pre-training data**  We examine the effect of the clustering level of pre-training data. Fig. 4 compares models pre-trained on datasets clustered at increasing sequence identity thresholds. ESM-1b is trained on sequences clustered at a 50% identity threshold. Improvements are seen using a 70% threshold with greatest improvement at 90%. Uniref100 performance appears to deteriorate early in training despite being the largest of the datasets. These results establish a link between model performance and the data distribution, highlighting the importance of training data in the design of protein language models.

**Scoring methods**  We compare four scoring methods on the validation set - masked marginals, wildtype marginals, mutant marginals, and psuedolikelihood. Table 5 shows that the masked marginal approach described in Eq. (1) outperforms other scoring methods, including ones in which the likelihood changes at non-mutated positions are considered. The scoring methods are described in detail in Appendix A.

**Parameter count**  Previous work with protein language models has established a link between model scale and learning of protein structure [12, 26]. We examine zero-shot transfer performance as a function of parameter count. We train models using the same width, depth, and learning rate as described in Henighan et al. [27], observing improvements with scale (Fig. 9). These findings suggest that continued scaling of the models will further improve results.

### 4.3 MSA Transformer

We examine how the sequences provided to MSA Transformer affect few-shot transfer. Table 8 compares sequence selection methods that vary the diversity of the sequences. Providing a more diverse set of sequences improves few-shot performance. Selecting a set of sequences to maximize diversity outperforms selecting a diversity minimizing set of sequences. Random sampling performs even better, and sampling sequences according to sequence weights [28] performs best.

We also vary the number of sequences used for inference. Fig. 11 shows few-shot performance as a function of the number of sequences given as input. The model performs well using only a few sequences, but performs best with 384 total sequences. In the main tables we report results sampling 384 sequences using sequence reweighting and ensembling predictions over five different subsamples from the MSA.

### 4.4 Unsupervised fine-tuning on MSAs

While ESM-1v performs well when evaluated in the zero-shot setting, we explore whether results can be improved by fine-tuning on the MSA. Fine-tuning on MSAs has been used in previous work [21, 16] as a stage in transfer learning to specialize a pre-trained model to a protein family, before applying supervision with labeled data. Here we consider using the fine-tuned model to make unsupervised predictions directly, without adding supervision from experimental data.

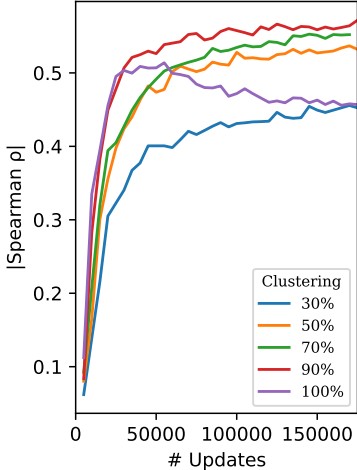

Figure 4: Comparison of pre-training datasets. Average |Spearman $\rho$| on the single-mutation validation set. While a 50% clustering threshold was used for ESM-1b, training with 90% clustering results in a significant improvement on variant prediction tasks. Notably, models trained on Uniref100, the largest dataset in this figure, appear to deteriorate early in training. These results establish a link between model performance and the data distribution, and highlight the importance of training data in the design of protein language models.

We observe that naively fine-tuning the model on the MSA results in rapid overfitting and poor performance on the prediction tasks (Fig. 12). While we experiment with a variety of approaches to freezing parameters during fine-tuning, detailed in Appendix B, none produce significant improvements. We find that an approach using pre-training sequences to regularize the fine-tuning performs well and enables training of all parameters without overfitting (Fig. 13). Spiked fine-tuning improves average absolute Spearman rho on the full dataset from 0.510 for zero-shot evaluation to 0.537 with fine-tuning.

## 5 Analysis of models

**Protein structure and function**    ESM-1v probabilities reflect the functional properties of sites within the protein. We use the entropy of the model's predictions for a position as a measure of its estimation of conservation. The lowest entropy predictions cluster at binding sites. Fig. 14 compares the distribution of the model's entropy between binding sites and non-binding sites. A significant difference is observed between the entropy assignment to binding and non-binding site residues. Fig. 5 visualizes the side chains of the 10 lowest entropy residues as predicted by the model on the crystal structure of DNA methyltransferase M.HaeIII interacting with its DNA substrate. In the crystal structure a cytosine of the substrate is inserted into the active site of the enzyme. The low entropy residues cluster in the active site and interact with the cytosine. Additional examples are visualized in Fig. 18.

The model probabilities also correspond to structure. Fig. 15 compares the entropy assigned to sites that are buried in the core of the protein vs. exposed on the surface. The model assigns significantly lower entropy to sites that are in the core of the protein, consistent with the idea that tight packing in the core places greater constraints on the selection of residues. Fig. 5B visualizes the entropy assigned by the model to each position overlayed on the structure of Indole-3-glycerolphosphate Synthase, a TIM barrel protein. Higher entropy is assigned to residues having outward facing side chains on the alpha helices, while lower entropy is assigned to the inward facing positions. Fig. 17 compares the probability assigned to hydrophobic, polar, and charged amino acids for buried sites vs. non-buried sites. The model prefers hydrophobic residues in the core and hydrophilic residues on the surface. The model probabilities closely match the empirical probabilities and those from the PSSM. Fig. 5C visualizes probability assigned to hydrophobic amino acids on the structure of

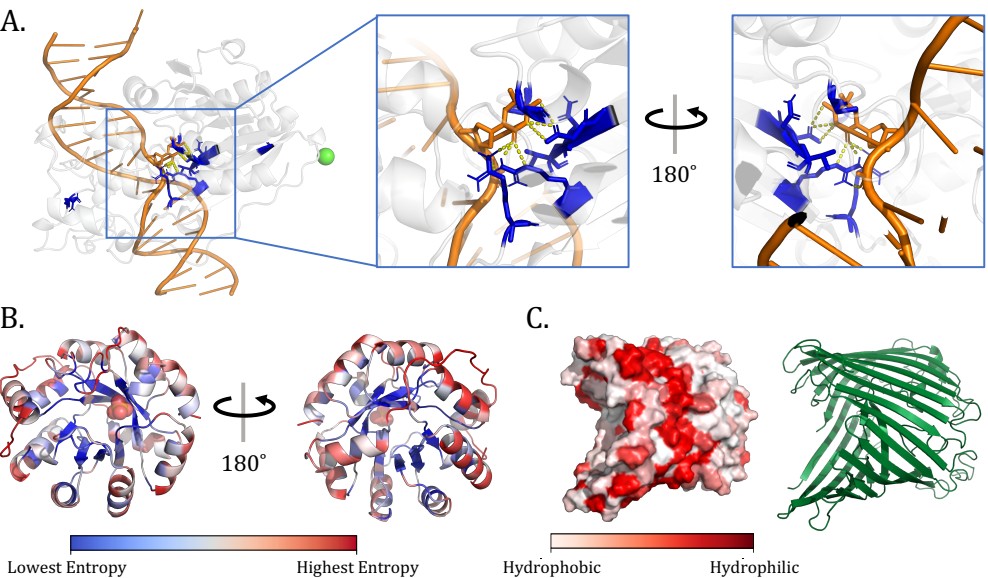

Figure 5: ESM-1v reflects the molecular basis of function in proteins. **(A)** DNA methylase HaeIII (pdbid: 1DCT [29]). Side chains for the top 10 positions with lowest prediction entropy shown in blue. Low-entropy positions cluster in the active site. **(B)** TIM Barrel (pdbid: 1IGS [30]) with residues colored by entropy. The model's predictions for residues on the surface have highest entropy (red) while those in the core have lower entropy (blue). Notably, residues on the alpha helices show a clear gradient from high to low entropy as residues transition from surface-facing to core-facing. **(C)** Sucrose-specific Porin (pdbid: 1A0T [31]), a transmembrane protein. The model predicts a hydrophobic band where the protein is embedded in the membrane.

Sucrose-specific Porin, a transmembrane protein. The model predicts a hydrophobic band in the center where the protein embeds in the membrane.

**Calibration**   We evaluate model calibration using 15008 sequences with length $< 1024$ from the trRosetta [32] dataset. ESM-1v probabilities for each amino acid at each position are calculated with the masked marginal probability in Eq. (1). Fig. 6 shows that the model is generally well calibrated for all amino acids except Methionine. ESM-1v always predicts Methionine as the first position in the sequence since full protein sequences always start with it, so care must be used when applying the model to subsequences. When excepting the first residue, the model achieves an average calibration error (defined for the multi-class setting in Appendix D.4) of 0.006.

We also explore the relationship between conservation (entropy of the PSSM) and the model's predicted entropy. Fig. 16 shows that these are well correlated (Pearson's $r = 0.44$), suggesting the model is able to identify conserved positions.

## 6   Related Work

### 6.1   Protein language models

In the past few years, a number of groups have developed language models for protein sequences [21, 33, 14, 34, 35, 22, 13, 12]. These models have been used for many tasks, including supervised low-N function prediction [16, 12], remote homology detection [22, 12], and protein generation [35]. The approach to the tasks typically involves transfer learning, where a pretrained language model is fine-tuned for a particular problem. Vig et al. [36] and Rao et al. [26] found that transformer attention corresponds to known biological properties such as structure and binding sites and can be used to predict contacts.

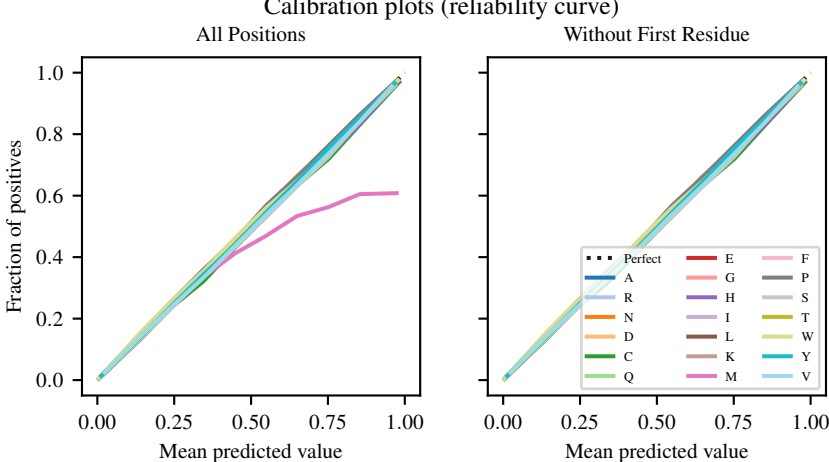

Figure 6: Calibration plot for ESM-1v predictions on each of the 20 naturally occurring amino acids on the trRosetta dataset. The multi-class classification is converted into a set of 20 one-versus-all classifications for the purpose of this analysis. Left and right plots show calibration of all positions and positions excluding the first residue, respectively. Since full sequences always start with Methionine, the model overwhelmingly predicts it in the first position. When evaluating the model on subsequences, such as those in the trRosetta dataset, this causes a miscalibration at the first residue. Including the first residue, the model has an average calibration error (ACE) of 0.011 in the first case and 0.006 in the second.

## 6.2 Mutation effect prediction

Supervised and unsupervised methods have been developed for prediction of mutational effects. Supervised methods train models using experimental measurements or labels from databases of clinical variants. Standard machine learning tools including linear regression, random forests, and support-vector machines can be used [37]. Models have been designed specifically for proteins, using feature engineering such as Envision [38] and PolyPhen-2 [39], ensemble methods such as Revel [40], MPC [41], CADD [42], and M-CAP [43], language models such as UniRep [21, 16] and ESM [12], and other representation learning approaches [44, 45].

Unsupervised mutation effect predictors work by inferring the likelihood of a mutation from the evolutionary landscape of the original protein. A density model fit to related sequences is used for scoring. SIFT [46] is a first order approach using a position-specific-scoring-matrix. EVMutation [4] extends this to a second-order approach by training a Potts model on the MSA. DeepSequence [20] includes higher-order interactions by training a VAE on the MSA instead, using the ELBO to score mutations. Riesselman et al. [47] proposes using an autoregressive model that does not require the sequences to be aligned. Laine et al. [48] uses an evolutionary tree structure inferred from the MSA to compare mutations.

Hsu et al. [15] show that unsupervised mutational effect predictors can be extended to perform supervised predictions, with better unsupervised predictors generally resulting in better supervised predictors. This suggests improving unsupervised prediction can drive progress in both settings. Concurrent with our work, Hie et al. [49] use open-source protein language models ESM-1b and TAPE to predict the direction of evolution in protein fitness landscapes.

## 7 Discussion

Advances in language modeling at scale are bringing the goal of a general purpose model for proteins closer to realization. This line of work aspires to a model that learns to read and write biology in its native language, that can be directly applied across a range of protein understanding and design tasks. For scalability, learning from sequences is important: while there are no central databases of high-throughput functional measurements, and few compilations exist, billions of sequences are

available to learn from in sequence databases [50, 51]. Sequences give an unparalleled view into the vast diversity and complexity of molecular parts invented by nature through billions of years of evolution.

Unsupervised structure [52–54, 28, 55, 56] and function [3, 4] learning methods first effectively realized the idea that biological properties could be read directly from sequences without supervision from experimental measurements. However these methods are not general purpose in the sense that a specialized model must be trained for every protein for which a prediction is to be made. We show that the same performance can be realized by a general purpose model that has been trained across many diverse protein families. Similar to observations on the learning of tertiary protein structure in large language models [12, 26], we find that increasing the scale of models leads to improvements in function learning. The understanding of mutational landscapes in the models correlates with the molecular basis of function in proteins, capturing binding sites and amino acid preferences that are determined by the folded structure.

Zero-shot transfer is an interesting capability of large scale language models, and represents a major point of departure from the unsupervised learning methods that are the basis for current state-of-the-art inference of protein structure and function. The capability for zero-shot transfer implies that a model can be trained once and then applied to perform inference for many tasks. It is also a window into deeper questions about the forms of generalization that are possible in learning from sequences. Reading structural and functional design principles from sequences is a necessary capability for writing new biologically active sequences. Generalization in the zero-shot setting suggests the potential for large language models to capture knowledge that can be transferred to generating new functional proteins.

## Acknowledgments and Disclosure of Funding

We thank Naman Goyal, Chloe Hsu, Zeming Lin, Ishan Misra, Myle Ott, Sergey Ovchinnikov, and Ethan Perez for valuable input on the paper. Roshan Rao was supported by funding from Facebook, Berkeley Deep Drive, and DARPA XAI.

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
