# A    Extraction methods

ESM-1v is pre-trained to output the probability for each possible amino acid at a masked position. We explore four methods of scoring the effects of mutations using the model:

- **Masked marginal**: Probabilities are extracted according to the mask noise during pre-training. At each position, we introduce a mask token and record the model's predicted probabilities of the tokens at that position.

- **Mutant marginal**: Probabilities are extracted according to the random token noise during pre-training. Among the 15% predicted positions in the sequence during pre-training, 10% of those are randomly mutated and 10% retain their original identities. The model is tasked to predict the correct token at those positions. Therefore, in this extraction method, we follow the pre-training methodology by passing in mutated tokens and recording the model's probability that they are correct.

- **Wildtype marginal**: We perform a single forward pass using the wildtype sequence. This method enables fast scoring as just a single forward pass is used.

- **Pseudolikelihood**: This method is proposed in the literature for scoring with masked language models [87].

In all cases, we assume an additive model when multiple mutations are present in a sequence. Results are summarized in Tables 5 and 7.

| Approach | Formal setting | Task level supervision | Representative Methods |
|---|---|---|---|
| Supervised mutation prediction | Supervised | Direct supervision from experimental measurements | Reviewed in [88] |
| Model trained on sequences from individual family | Unsupervised | Weak positive supervision from MSA | [4, 20] |
| Fine-tuning on experimental data | Semi-supervised transfer | Supervision from experimental measurements | [21, 16, 12] |
| Fine-tuning on MSA | Transfer learning with weak-positive supervision | Weak positive supervision from MSA | Introduced for transfer learning in [21] |
| Direct forward pass | Zero-shot learning | None | This work |

Table 3: Zero-shot learning is a natural extension of the various approaches that have been used for mutational effect prediction to date. Rather than training a new model for every task, a single general purpose model is trained and can be directly applied across multiple tasks. The approach is fully unsupervised, no information from experimental measurements of function is used.

Let $x^{mt}$ and $x^{wt}$ represent the mutant and wildtype sequences. We refer to $x_{-i}$ as the sequence $x$ with a mask introduced at position $i$. We refer to the set of mutations that are introduced as the set $M$. For example, if mutations are introduced at positions 3 and 6, then $M = \{3, 6\}$.

**Masked marginal probability (L forward passes)**    This method performs best among the four. We introduce masks at the mutated positions and compute the score for a mutation by considering its probability relative to the wildtype amino acid (Strategy a):

$$\sum_{i \in M} \log p(x_i = x_i^{mt} | x_{-M}) - \log p(x_i = x_i^{wt} | x_{-M})$$

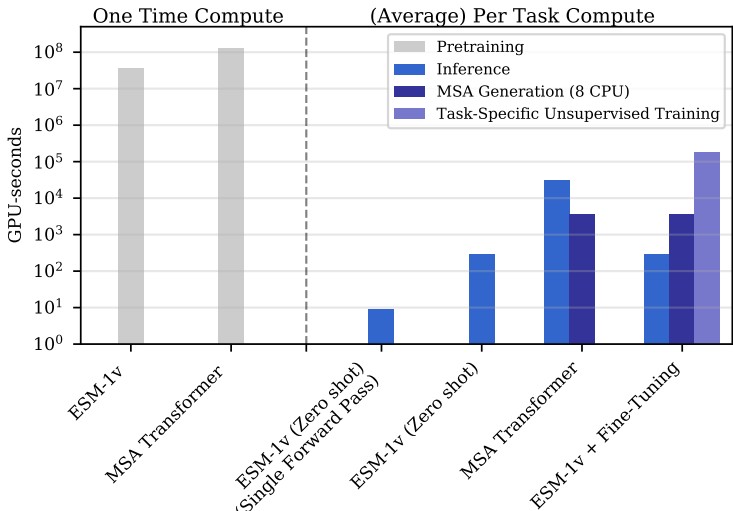

Figure 7: Compute requirements in GPU-seconds for (left) pre-training and (right) average task. With open-sourced pre-trained models, end users bypass the pre-training phase and only incur inference costs. ESM-1v and MSA Transformer amortize compute cost into a single expensive pre-training run. After pre-training, inference is fast. On average, it takes 10 seconds to label a deep mutational scan from Riesselman et al. [20] with ESM-1v (Zero-shot, Single Forward Pass). Performance improves marginally with the more expensive scoring scheme (Table 5).

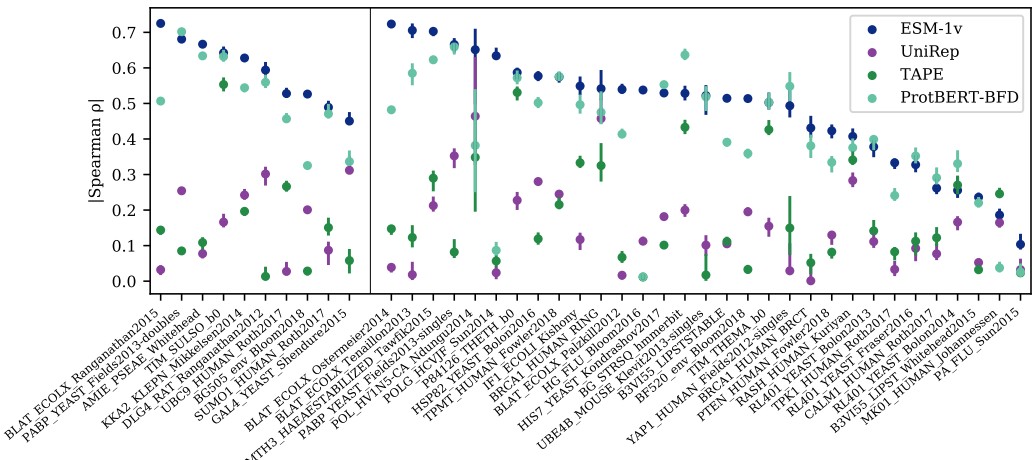

Figure 8: Zero-shot performance of ESM-1v compared to earlier protein language models on all 41 deep mutational scans. Points are |Spearman $\rho$| on each dataset, error bars show standard deviation of 20 bootstrapped samples. Validation proteins are shown to the left of the dividing line and test proteins to the right. ESM-1v is the best performing method on 30 of the 41 deep mutational scans.

This formulation assumes an additive model, consistent with the training objective. We show that this assumption is justified empirically by evaluating the model with different choices at the non-mutated positions. First, the wildtype sequence (Strategy b):

$$\sum_{i \in M} \log p(x_i = x_i^{mt} | x_{-i}^{mt}) - \log p(x_i = x_i^{wt} | x_{-i}^{wt})$$

and the mutant sequence (Strategy c):

$$\sum_{i \in M} \log p(x_i = x_i^{mt} | x_{-i}^{mt}) - \log p(x_i = x_i^{wt} | x_{-i}^{mt})$$

| Clustering | \| Spearman $\rho$ \| |
|---|---|
| 30% | 0.456 |
| 50% | 0.537 |
| 70% | 0.552 |
| 90% | **0.564** |
| 100% | 0.458 |

Table 4: Average |Spearman $\rho$| on the single-mutation validation set after training a 650M parameter Transformer model for 170,000 updates on various sequence identity clusterings of Uniref.

| Method | \| Spearman $\rho$ \| |
|---|---|
| Masked marginal | **0.582** |
| Mutant marginal | 0.578 |
| Wildtype marginal | 0.572 |
| Pseudo-likelihood | 0.552 |

Table 5: Benchmarking scoring schemes on the single-mutation validation set. The means across the validation set are listed. The masked marginal scheme performs best.

Strategy (a), where we mask all positions at the same time, performs best on the PABP Yeast Doubles validation dataset (Table 7).

**Mutant marginal probability**   This method is analogous to the wildtype marginal probability, except we use the mutant sequence instead.

$$\sum_{i \in M} [\log p(x_i = x_i^{mt}|x^{mt}) - \log p(x_i = x_i^{wt}|x^{mt})]$$

This method requires a single forward pass for every mutation.

**Wildtype marginal probability (1 forward pass)**   In the fastest scheme, we perform a single forward pass using the wildtype sequence as input. For a set of mutations at positions $M$, the score is:

$$\sum_{i \in M} [\log p(x_i = x_i^{mt}|x^{wt}) - \log p(x_i = x_i^{wt}|x^{wt})]$$

We find that the method performs well with a minor 1% decrease in absolute performance, while requiring very limited computational resources. The strong performance indicates that the masked language modeling objective causes the model to capture the fitness landscape of the protein in its outputs.

**Pseudolikelihood**   Psuedolikelihood has been proposed in the literature as a method to score sequences using masked language models [87]. We compute the score as follows:

$$\sum_{i} \log p(x_i = x_i^{mt}|x_{-i}^{mt}) - \log p(x_i = x_i^{wt}|x_{-i}^{wt})$$

As mutation prediction is a ranking task and as the contribution from the second term is constant throughout the deep mutation scan (i.e. the wildtype sequence is always the same), we can safely drop it from the computation.

## A.1   Evaluation

We compare the methods described above on the validation set, finding that the masked marginal scheme performs best. To determine the specific mode of inference when multiple mutations are present, we examine each method on the "doubles" component of the PABP Yeast dataset finding the masked marginal (a) strategy performs best. This scoring method is used across the results.

| Input | Consensus columns only | \| Spearman $\rho$ \| |
|-------|------------------------|----------------------|
| MSA seed | Yes | 0.573 |
| MSA seed | No | 0.567 |
| Uniprot | N/A | **0.582** |

Table 6: ESM-1v performs better when including the full protein sequence as listed in Uniprot, compared to using the seed sequence of the MSA corresponding to the deep mutational scan. Results on single-mutation validation set. The means across the validation set are listed. We experiment with a number of strategies for inference: (i) the consensus columns only; (ii) the aligned part of the query sequence; and (iii) the complete Uniprot sequence. The complete Uniprot sequence performs best, possibly because the model was pre-trained on complete Uniprot sequences. We use the MSA seed sequence from the MSAs released by [20] corresponding to the deep mutational scans.

| Method | \| Spearman $\rho$ \| |
|--------|----------------------|
| Masked marginal (a) | **0.692** |
| Masked marginal (b) | 0.482 |
| Masked marginal (c) | 0.483 |
| Mutant marginal | 0.694 |
| Wildtype marginal | 0.672 |
| Pseudo-likelihood | 0.608 |

Table 7: Ablating scoring schemes on the PABP Yeast Doubles dataset. The masked marginal scheme performs best when masking all mutated sites together. Mean absolute Spearman $\rho$ across the single-mutation validation tasks is reported.

## A.2 Evaluating ESM-1v on subsequences

DeepSequence, EVMutation, and the MSA Transformer use the consensus columns of a MSA as input. We construct MSAs using the seed sequences from the DeepSequence paper, which usually correspond to a subsequence of the protein capturing the domain where the deep mutational scan was performed.

Table 6 explores using the MSA seed sequence vs. the full Uniprot sequence for inference on the validation set. We find that the full Uniprot sequence performs best, possibly because the model was pre-trained on Uniprot sequences. We note in Figure Fig. 6 that the model captures some bias in the Uniprot dataset, for example that most proteins begin with a methionine (corresponding to the start codon).

## B Unsupervised fine-tuning ESM-1v

**Experimental setup** We assess a number of approaches for fine-tuning ESM-1v on task-specific MSAs. We evaluate modeling decisions by fine-tuning on tasks from the validation set and examining the mean change in Spearman $\rho$ over the course of training. For efficiency, we compute Spearman $\rho$ using the wildtype marginal strategy, as this requires just a single forward pass. After the final modeling decisions are selected, we train all models for 7500 updates and evaluate on all proteins using the masked marginals strategy. All models in this section were trained with a constant learning rate of $10^{-5}$ using the masked language modeling objective. For reference, the ESM-1v pre-training was performed with a target batch size of 1M tokens.

**Unsupervised fine-tuning baselines** The concept of unsupervised fine-tuning of an MSA has been previously proposed [21, 16]. Fig. 12 studies a basic fine-tuning setup on the consensus columns of the MSA. Each model is fine-tuned on a single MSA with a target batch size of 8192 tokens. We first observe that that models overfit quickly if the entire model is trained. This results in a decrease in Spearman $\rho$ compared to initialization. As the fine-tuning is performed on the consensus columns of the MSA, we sought to regularize the model by fine-tuning only the embeddings. As a PSSM already captures information relevant to the task, we hypothesize that tuning the embeddings could

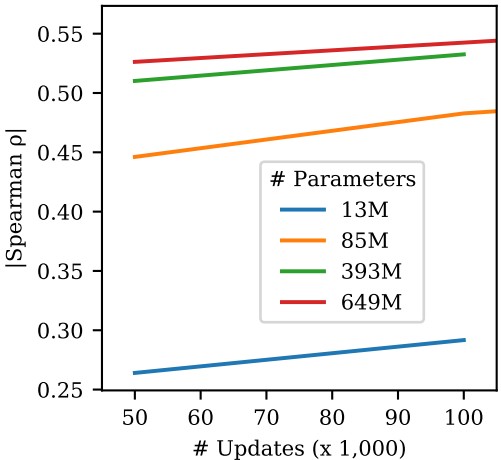

Figure 9: Larger models perform better on variant prediction. We trained four models of various scales, following the hyperparameters listed in Henighan et al. [27]. Results on single-mutation validation set.

| MSA Subsample Strategy | Context size | \| Spearman $\rho$ \| |
|---|---|---|
| Diversity minimizing | 256 sequences | 0.255 |
| Random | 256 sequences | $0.535 \pm 0.024$ |
| HHFilter | 256 sequences | $0.550 \pm 0.015$ |
| Sequence reweighting | 256 sequences | $0.578 \pm 0.005$ |

Table 8: Subsampling strategies for MSA Transformer evaluated on the single-mutation validation set. Sequence reweighting performs best. When sampling methods are stochastic, 5 seeds are run and the mean and standard deviation is reported. With HHFilter, we run with the `-diff M` parameter and randomly subsample the output if more than `M` sequences are returned. We use a coverage parameter of 75 and a sequence identity parameter of 99. Mean absolute Spearman $\rho$ across the single-mutation validation tasks is reported.

capture similar information and boost performance. Similarly, we experiment with tuning only the layer normalizations, as these have also been recently shown to enable transfer to new tasks. In both cases, we found no improvement to the average Spearman $\rho$. We also assessed label smoothing and replacing the gap token with a mask token or a pad token finding no significant impact; for simplicity, we omit label smoothing and use the mask token for future experiments.

**Minimal models**   We also examine a set of minimal models, in which we freeze all parameters in the Transformer and learn a projection from the ESM-1v outputs onto a PSSM, taking the sum of the projection and the PSSM. We experiment with freezing the PSSM or allowing it to train. We did not see a change in Spearman $\rho$ of more than 0.01.

**Spiked unsupervised fine-tuning**   Next, we examine a new strategy, which we call `spiked fine-tuning`. In spiked fine-tuning, we regularize the fine-tuning by continuing to spike pre-training sequences into the fine-tuning batch. In this setting, we train on the entire MSA, including non-consensus positions. We find that spiked fine-tuning with a small ratio (0.01) of MSA tokens to pre-training tokens performs best and enables training of all parameters without overfitting.

The final models were trained for 7500 updates using spiked fine-tuning with a batch size of 500k tokens. To produce an ensemble, we perform the fine-tuning scheme on five models that were pre-trained with different seeds. Each model was also fine-tuned with a unique seed.

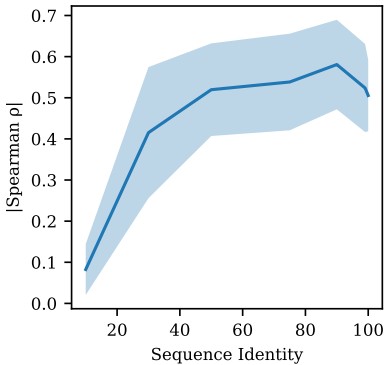

Figure 10: Filtering sequences with high sequence identity to the query improves performance. The curve illustrates mean ± standard deviation across the 9 validation proteins. HHFilter is used to filter the MSAs with coverage of 75 and various sequence identity values as shown on x-axis. After filtering, 384 sequences are sampled for inference. Each sequence identity value $s$ refers to using sequences with no more than $s\%$ sequence identity to the seed sequence. The MSA Transformer appears to primarily use sequences that are close to the seed sequence, yet performance drops if sequences that are *too similar* remain in the MSA. Results are broken down across the single-mutation validation set in Table 9.

| Sequence Identity (%) | 10 | 30 | 50 | 75 | 90 | 99 | 100 |
|---|---|---|---|---|---|---|---|
| AMIE_PSEAE_Whitehead | 0.025 | 0.461 | 0.467 | 0.365 | **0.665** | 0.654 | 0.622 |
| BG505_env_Bloom2018 | 0.055 | 0.055 | 0.417 | **0.482** | 0.452 | 0.450 | 0.457 |
| BLAT_ECOLX_Ranganathan2015 | 0.060 | 0.630 | 0.745 | 0.776 | **0.795** | 0.662 | 0.478 |
| DLG4_RAT_Ranganathan2012 | 0.008 | 0.431 | 0.416 | 0.431 | **0.457** | 0.418 | 0.400 |
| GAL4_YEAST_Shendure2015 | 0.080 | 0.287 | 0.366 | 0.441 | **0.576** | 0.542 | 0.388 |
| SUMO1_HUMAN_Roth2017 | 0.131 | 0.430 | 0.516 | **0.541** | 0.500 | 0.492 | 0.495 |
| TIM_SULSO_b0 | 0.026 | 0.581 | 0.633 | 0.625 | **0.649** | 0.324 | 0.632 |
| UBC9_HUMAN_Roth2017 | 0.165 | 0.376 | 0.557 | **0.583** | 0.494 | 0.555 | 0.475 |
| KKA2_KLEPN_Mikkelsen2014 | 0.192 | 0.484 | 0.560 | 0.601 | **0.637** | 0.616 | 0.602 |

Table 9: Filtering MSAs from the single-mutation validation set with HHFilter coverage 75 and various sequence identity values. Filtering sequences to an identity threshold of 75% or 90% consistently performs best. The Spearman rank correlation between MSA Transformer predictions and experimental data is shown for each deep mutational scan.

## C   Datasets

### C.1   Evaluation Tasks

We evaluate models on a set of 41 deep mutational scans collected by Riesselman et al. [20], which comprise a variety of tasks assessing a diverse set of proteins. Across tasks, the experimental data differ widely in the functions tested and in the experimental measurements performed. Of the 41 datasets, 37 are single-mutation only, 1 is double-mutation only, and the rest contain a variable number of mutations per sequence between 1 and 28. The median number of mutations is 2979, and the average is 16822; the smallest dataset has 37 mutations, and the largest has 496137. We randomly select 9 single-mutation experiments as a validation set. We also ablate the multiple mutation scoring approach on the double mutations from the PABP Yeast deep mutational scan. We exclude the 10 tasks used for validation and ablations from the test set. These datasets are reported in results for the full set. While the original compilation has 43 datasets, we exclude the tRNA (which is not a protein) and the toxin-antitoxin complex (which comprises multiple proteins).

We treat each deep mutational scanning dataset as a separate prediction task, scoring each of the variants in the dataset with the model. We evaluate performance by comparing the scores with the experimental measurements using Spearman rank correlation. Results are broken out between the

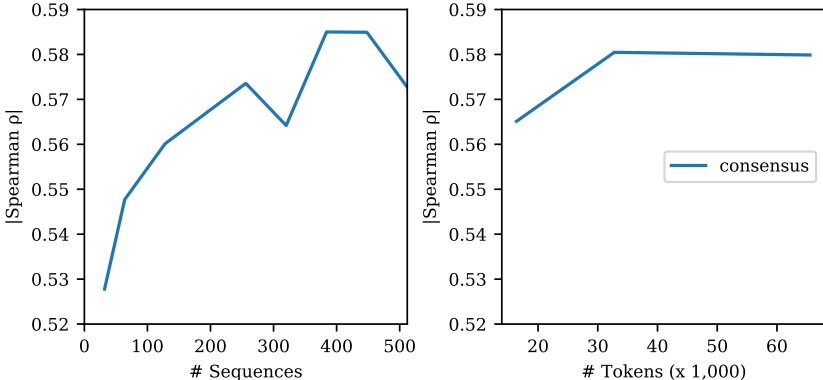

Figure 11: Few-shot performance of the MSA Transformer is robust to the number of sequences used for inference. **Left:** Varying the number of sequences used in inference. **Right:** Varying the number of tokens used for inference. Since the number of sequences in each MSA varies, we assess the effect of fixing the total number of tokens sampled from each MSA and drawing the corresponding number of sequences to fill the context. Results on single-mutation validation set.

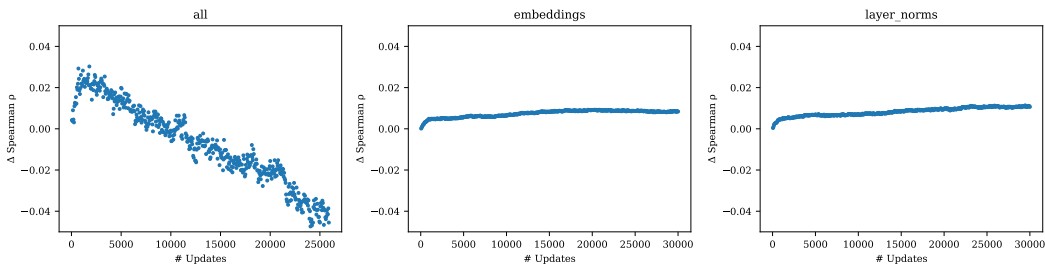

Figure 12: Unsupervised fine-tuning baselines. Mean change in Spearman $\rho$ across 9 models trained on the single-mutation validation set tasks. The title of each plot denotes the parameters that are trained. We find that fine-tuning the entire model results in overfitting, but limiting the training to just the embeddings or just the layer norms does not improve performance with respect to the pre-trained initialization. The choice of gap token and label smoothing has limited effect.

test set, which excludes the validation set, as well as the full set of 41 datasets. All ablations are performed on the single mutant validation set or the PABP Yeast doubles experiment. Only the final models are evaluated on the test set.

## C.2 Pre-training datasets

For the clustering sweep in Fig. 4, we use the Uniref50 and Uniref90 databases from the `2020_03` release of Uniref [24], a publicly available database of proteins, clustered respectively to 50% and 90% sequence identity. For the 30% sequence identity dataset, we use Uniclust30 `2020_03` [90]. For the 70% sequence identity dataset, we Uniref100 is hierarchically clustered to the 90%, then 70% sequence identity level. MMseqs settings are those used by Uniref: 80% overlap with longest sequnece in the cluster, which translates to `mmseqs-cluster -min-seq-id 90,70 -cov-mode 0 -alignment-mode 3 -c 0.8`. In order to compute pre-training perplexities on a heldout validation set, we randomly select 1% of sequences each from Uniref30, Uniref50, and Uniref90. We then exclude sequences that are similar to the validation sequences by removing all sequences found with MMSeqs search (`-min-seq-id 0.xx`) for validation set xx. We use the most sensitive settings in MMSeqs `-alignment-mode 3 -max-seqs 300 -s 7`, taking the train set as the query database and the validation set as the target database. We use settings `-c 0.8 -cov-mode 0` to match the settings of Uniref. Pretraining perplexities on the validation sets are reported in Table 10.

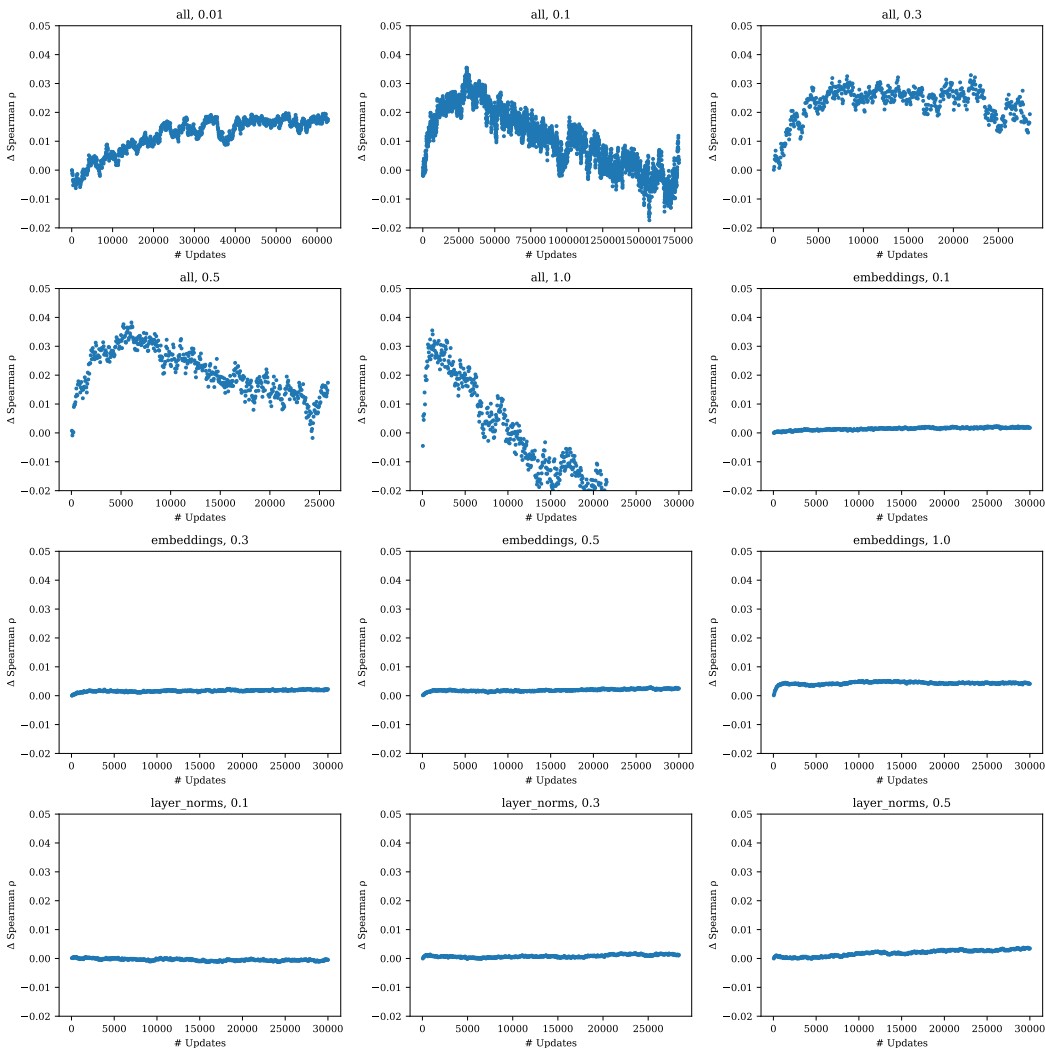

Figure 13: Spiked unsupervised fine-tuning. Mean change in Spearman $\rho$ across 9 models trained on the single-mutation validation tasks. The title of each plot denotes the parameters that are trained; and the ratio of MSA tokens to pre-training tokens. We find that a small ratio performs well and reduces the tendency for the model to overfit, while preserving strong performance. Performance is not improved if the fine-tuning is limited to just the embeddings or just the layer norms.

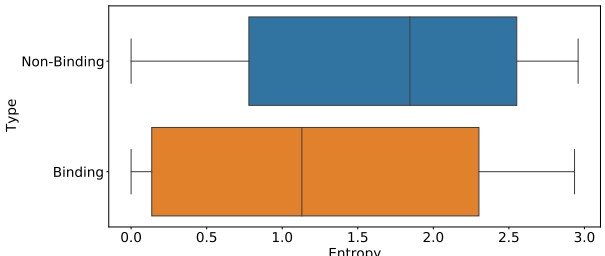

Figure 14: Box plot comparing entropy scores for binding vs non-binding positions in structures labeled in the Provis validation dataset (as described in Appendix B.4 of [36]). A Welch's *t*-test determines that the difference between the two means is statistically significant (p < 0.01).

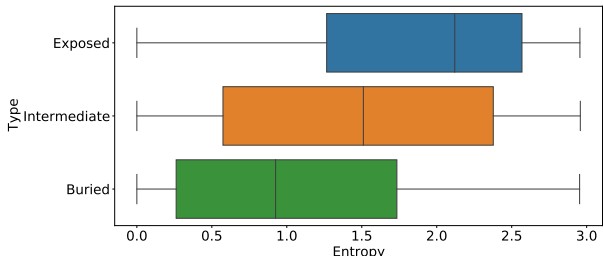

Figure 15: Box plot comparing entropy scores across residue depths in structures from the trRosetta dataset. Residue depths are categorized based on the number of neighboring residues with C-beta distance<10 angstroms. (exposed≤16, buried≥24 [89]). A one way Anova test determines that the differences between all three means are statistically significant (p < 0.01).

## C.3 Baselines

The MSAs used for training DeepSequence and EVMutation are generated from the 2017-10 version of Uniref100, whereas the models we study are trained on sequences from Uniref90 2020-03. In the case of MSA Transformer, the model is pre-trained on the 2018-03 Uniref, but we use 2020-03 MSAs for inference. In order to provide a fair comparison, we regenerate MSAs against the 2020-03 Uniref according to the methodology in Hopf et al. [4] and retrain EVMutation (replication) and DeepSequence (replication) on these datasets using their open-source codebases. For the viral proteins BF520_env_Bloom2018, BG505_env_Bloom2018, HG_FLU_Bloom2016, PA_FLU_Sun2015, POLG_HCVJF_Sun2014, POL_HV1N5-CA_Ndungu2014, we compute the sequence weights with $\theta = 0.01$ (versus default $\theta = 0.2$) following Riesselman et al. [20]. In the replication of the DeepSequence ensemble, for BF520_env_Bloom2018, BG505_env_Bloom2018, one of the five runs failed so we reran with a different random seed.

## C.4 Validation and test set

The single-mutation validation set consists of the following deep mutational scans: AMIE_PSEAE_Whitehead, BG505_env_Bloom2018, BLAT_ECOLX_Ranganathan2015, BRCA1_HUMAN_RING, DLG4_RAT_Ranganathan2012, GAL4_YEAST_Shendure2015, POLG_HCVJF_Sun2014, SUMO1_HUMAN_Roth2017, TIM_SULSO_b0, UBC9_HUMAN_Roth2017, KKA2_KLEPN_Mikkelsen2014.

For ablations studies with multiple mutations the following dataset is used: PABP_YEAST_Fields2013-doubles

The test set consists of the following deep mutational scans: B3VI55_LIPSTSTABLE, B3VI55_LIPST_Whitehead2015, BF520_env_Bloom2018, BG_STRSQ_hmmerbit, BLAT_ECOLX_Ostermeier2014, BLAT_ECOLX_Palzkill2012, BLAT_ECOLX_Tenaillon2013, BRCA1_HUMAN_BRCT, CALM1_HUMAN_Roth2017, HG_FLU_Bloom2016, HIS7_YEAST_Kondrashov2017, HSP82_YEAST_Bolon2016,

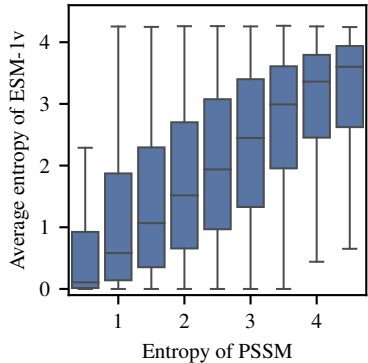

Figure 16: Entropy of PSSM versus ESM-1v predicted entropy on the trRosetta dataset. PSSM entropy determines the level of conservation at a given position in a protein family. ESM-1v entropy is well correlated with PSSM entropy (Pearson's $r = 0.44$), suggesting the model is able to identify conserved positions.

| Ground Truth | | | | ESM (Uniref90) | | | | PSSM | | |
|---|---|---|---|---|---|---|---|---|---|---|
| Exposed | 0.36 | 0.29 | 0.35 | | 0.38 | 0.29 | 0.33 | 0.39 | 0.29 | 0.32 |
| Intermediate | 0.55 | 0.28 | 0.17 | | 0.52 | 0.28 | 0.19 | 0.53 | 0.28 | 0.19 |
| Buried | 0.69 | 0.24 | 0.068 | | 0.67 | 0.25 | 0.088 | 0.67 | 0.24 | 0.086 |
| | Hydrophobic | Polar | Charged | | Hydrophobic | Polar | Charged | Hydrophobic | Polar | Charged |

Figure 17: Predicted distribution of hydrophobic, polar and charged amino acids at the surface and core of proteins in the trRosetta dataset. We compare to the actual proportion in the protein structure. We classify residues into buried, intermediate or exposed by residue depths based on the number of neighboring residues with C-beta distance $< 10$ angstroms (exposed $\leq 16$, buried $\geq 24$) [89]. ESM-1v and PSSM both see increased hydrophobicity predictions for buried residues, in correspondence with the ground truth data. Predicted probabilities are produced by introducing a mask token at each position.

```
IF1_ECOLI_Kishony, MK01_HUMAN_Johannessen, MTH3_HAEAESTABILIZED_Tawfik2015,
P84126_THETH_b0, PABP_YEAST_Fields2013-singles, PA_FLU_Sun2015,
POL_HV1N5-CA_Ndungu2014, PTEN_HUMAN_Fowler2018, RASH_HUMAN_Kuriyan,
RL401_YEAST_Bolon2013, RL401_YEAST_Bolon2014, RL401_YEAST_Fraser2016,
TIM_THEMA_b0, TPK1_HUMAN_Roth2017, TPMT_HUMAN_Fowler2018,
UBE4B_MOUSE_Klevit2013-singles, YAP1_HUMAN_Fields2012-singles.
```

# D    Methodology

## D.1    Model selection

ESM-1b and MSA Transformer model checkpoints are selected based on performance on the single mutation validation set. Open sourced checkpoints are used for ESM-1b and other protein language model baselines.

## D.2    Treatment of synonymous mutations

Synonymous mutations are mutations in DNA that do not change the protein sequence that is expressed. The deep mutational scanning datasets that we evaluate here can therefore include DNA

mutations that do not change the protein sequence itself. Synonymous mutations are excluded from results.

### D.3 Bootstraps

To compute bootstraps for the pointplots, we randomly resample each deep mutational scan (with replacement) and compute the Spearman $\rho$ between the experimental data and model predictions.

### D.4 Average calibration error

The standard expected calibration error (ECE) performs poorly for highly imbalanced data [91]. Following Neumann et al. [91] and Nixon et al. [92] we adapt average calibration error for the multi-class setting as follows:

$$\frac{1}{K} \sum_{k=1}^{K} \frac{1}{B_k^+} \sum_{b=1}^{B_k^+} |\texttt{acc}(b, k) - \texttt{conf}(b, k)|$$

where $K$ is the number of classes, $B_k^+$ is the number of non-empty bins for class $k$, and $\texttt{acc}$ and $\texttt{conf}$ are the accuracy and confidence for bin $b$ and class $k$.

## E  Performance by MSA depth

We examine the relationship between the number of related sequences in the pre-training set and performance on the task. We use Jackhmmer [93] version 3.3.1 with a bitscore threshold of 27 and 8 iterations to construct MSAs from the ESM-1v training set. We do not observe a strong correlation between MSA depth and the observed absolute value of Spearman $\rho$ (Figure Fig. 19).

## F  Compute costs

ESM-1v models are pre-trained for 6 days on 64 V100 GPUs. Weights for the MSA Transformer were retrieved from the open-source repository released by the authors; the model was pre-trained for 13 days on 128 V100 GPUs. Once trained, the models can be used directly for function prediction tasks. Forward inference is efficient, meaning that for applications of the models, the additional compute is minimal. In total, five ESM-1v models were trained on various Uniref clustering thresholds to five different levels: 30%, 50%, 70%, 90%, and 100%. For the 90% sequence identity level, five total models with different random seeds were trained, for use in an ensemble. As illustrated in Fig. 7, inference is inexpensive by comparison. Batch inference was performed with preemptible, short (shorter than one hour), single V100 GPU jobs on a shared compute cluster.

| Clustering | Valid (30%) | Valid (50%) | Valid (90%) |
|---|---|---|---|
| 30% | 8.93 | 8.33 | 7.29 |
| 50% | 8.90 | 7.77 | 6.27 |
| 70% | 9.05 | 7.80 | 5.85 |
| 90% | 9.37 | 8.10 | 5.56 |
| 100% | 9.89 | 8.65 | 6.05 |

Table 10: Perplexities on heldout pre-training validation sequences after training a 650M parameter Transformer model for 170,000 updates on various sequence identity clusterings of Uniref.

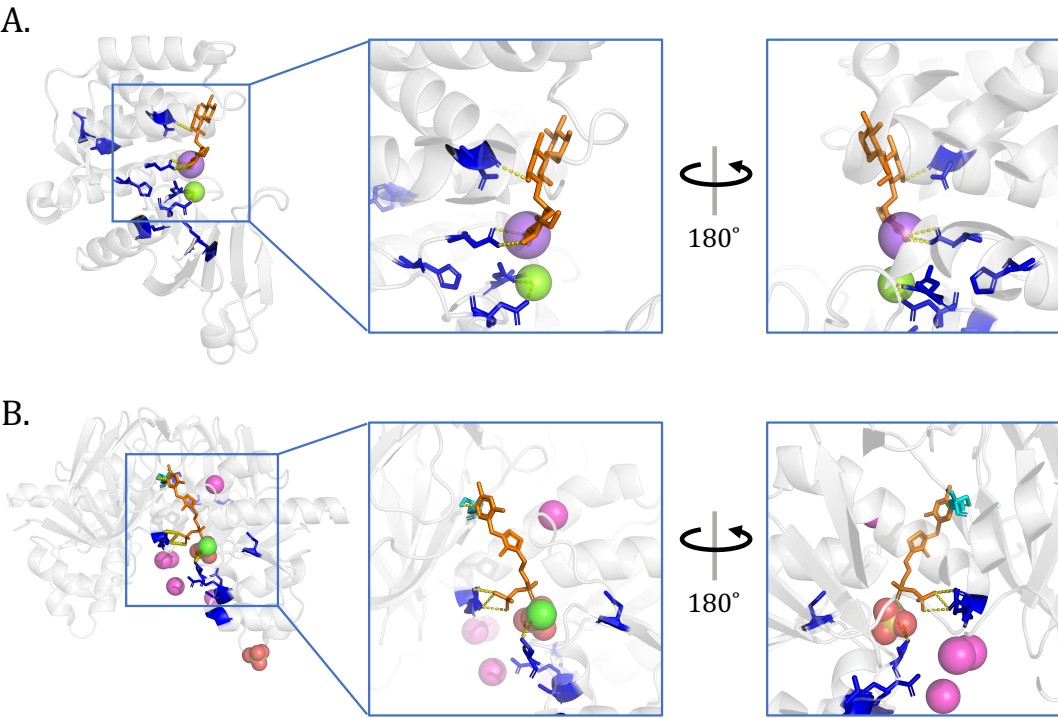

Figure 18: ESM-1v accurately captures functional properties. Further examples. The ten positions with lowest predicted entropy highlighted in blue. **(A)** Kanamycin kinase APH(3')-II (pdbid: 1ND4 [94]). The highlighted residues interact with the kanamycin aminoglycoside, as well as the magnesium and sodium ions. **(B)** Thiamin pyrophosphokinase 1 (pdbid: 3S4Y). Residue 216 is one of the 10 lowest entropy residues, and we highlight it on the other chain (in cyan) to show both chains of the dimer interacting with the thiamine diphosphate.

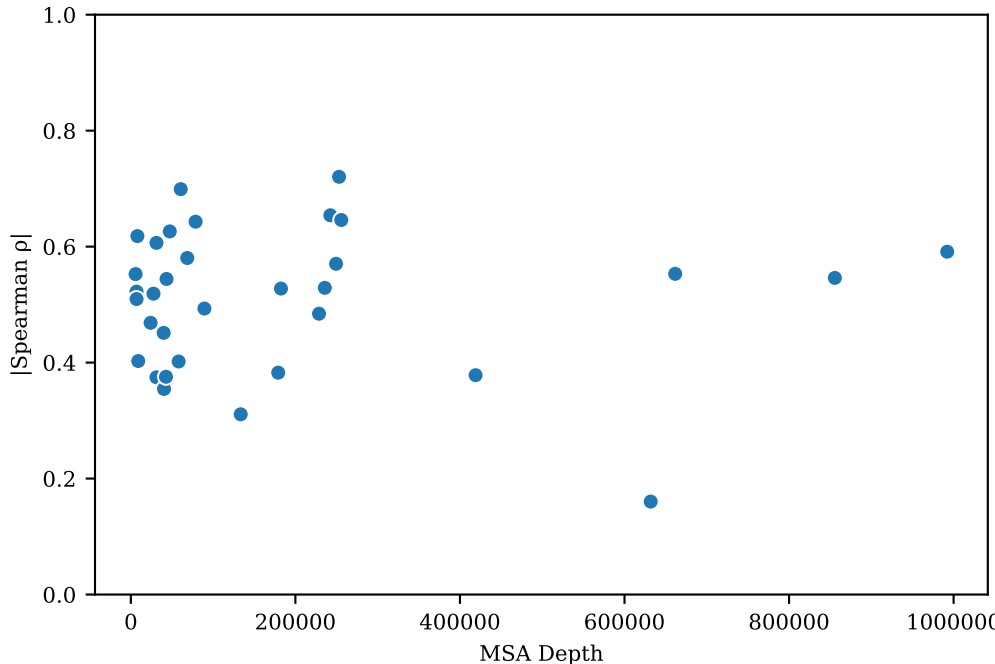

Figure 19: Relation between MSA depth and zero-shot performance of ESM-1v. We use JackHMMer [93] version 3.3.1 with a bitscore threshold of 27 and 8 iterations to construct MSAs from the ESM-1v training set. We do not observe a strong correlation between MSA depth and the observed |Spearman $\rho$|.