# OpenReview forum: "Language models enable zero-shot prediction of the effects of mutations on protein function"
_NeurIPS.cc/2021/Conference — NeurIPS 2021 Poster_

### Official Review · Reviewer_bbgP · 2021-07-06

**Rating:** 6
**Confidence:** 5

**Summary:**

The paper adds a follow-up experiment to a recent paper demonstrating the power of large language models (particularly, BERT and the MSA transformer) for a variety of protein structure and function prediction tasks. The new task (zero-shot scoring of the effects of mutations) has a variety of useful applications for clinical medicine and protein design and has been explored extensively in recent papers that used alternative models. This paper demonstrates that the these new models perform similarly, or perhaps slightly better, on a standard suite of datasets.





**Limitations And Societal Impact:**

Overall yes, however this statement should be revised: " Otherwise,basic methods in computational biology have a long time horizon to concrete social impact." Just because something has a long time horizon doesn't mean that it is benign.

**Main Review:**

**originality**

My primary negative critique of this paper is its lack of novelty.

It has been previously established in the literature by many papers that the functional effect of mutations can be predicted in a zero-shot setting by scoring variants using the likelihood of a generative model. Further, the two models considered in this paper have been introduced in recent prior work. Those papers are really great; I found them rigorous and well written and would have advocated for their acceptance to NeurIPS. I feel that the scale of contributions of the current paper is smaller than that of a typical NeurIPS paper, though, and its content would be better placed as an additional section of experiments in, for example, the MSA transformer paper.

One argument for a paradigm shift due to your approach is that practitioners don't have to train a new model for a protein family of interest and don't have to perform multiple sequence alignment of homologous sequences. Instead, the user can just do forward passes in an existing model. However, it's rare for practitioners to be considering a large number of families at once. See further comments below.

**clarity/quality**
The paper is well written and cites relevant recent work. The experimental results could have been analyzed in more detail. See comments below.

**significance**
This paper is part of a really important line of work on deep learning + proteins. My hesitation is around whether this paper should appear as a stand-alone paper at NeurIPS, not whether the results are of interest to the community.

**zero-shot prediction**

I disagree with your definition of zero-shot in the paragraph starting on line 75. To me, any method is zero-shot if it doesn't rely on experimental data as input. Why is a method that relies on querying a sequence database to retrieve homologous sequences qualitatively different than your approach?  Doesn't the training set for ESM-1v or MSATransformer include the same 'weak supervision' that the traditional approach relies on?

I suppose you could argue that your method could be applied to predict the functional effect of mutations on sequences that are very different than anything that appears in nature, where there is no associated protein family that could be retrieved by a database search. This is zero-shot in a different sense and not evaluated in the experiments.

**computing the likelihood of a variant**

In Eq (1) uses P(x_t | x_{-t}), but ignores P(x_{-t} | x_t). Why is this a reasonable assumption? Surely, when making a mutation at a particular position, it effects the likelihood of the values at the other positions too. Note that prior models for this task (VAEs, autoregressive models, Potts models) don't make such a simplifying assumption. I see that you discuss alternatives in Appendix A. Why do you think this simple approach is sufficient?

 **Epistasis**
Following up on the previous point, does this mean that you can't capture epistasis, which is known to be prevalent in fitness landscapes? For datasets with multiple mutations, you can compute the subset of examples that are epistatic (e.g., it the variant is non-functional, but combines mutations that led to a functional variant when used individually). On these, how do you compare to DeepSequence, which can capture epistasis?

**ESM-1b vs. ESM-1v**
Am I correct in understanding that ESM-1v is the same as ESM-1b, but trained on slightly different data? When I discuss originality above, I argue that the paper does not introduce a new model. To me, changing the training data does not amount to introducing a new model with the degree of originality characteristic of a stand-alone paper.


**Inference speed**
I don't feel that this is a particularly important distinction between the proposed model and prior work. It's not clear what an application setting would be where practitioners need to make predictions for many protein families at high throughput. If the goal is just to predict  the functional effects of mutations on a single parent sequence, then the speed difference across techniques will be minor.  The goal of this zero-shot prediction is to avoid needing to run wet-lab experiments, so any approach that can run on a computer in less than an hour is a massive speedup vs. the alternative.


 **Evaluation Metrics**
Can you please explain why spearman correlation is a reasonable performance metric? This is noisy experimental data. What do you think is the upper limit on performance? If you binarized the experiment measurement (e.g. whether a variants is worse than the wildtype or better), what would detection metrics like AUC or precision@k look like? I'm also assuming that such labels would be very imbalanced (with more than half of the mutations leading to a decrease in function). Doesn't that bias spearman rho, and make it less predictive of performance for protein design applications where the goal is to increase fitness?

**Break-down of performance**

I would have appreciated more analysis of where the method succeeds and fails. What characterizes datasets where your methods outperform/underperform prior work? For example, how do you compare on multi-mutant variants?

**Sec 5**
Again, this feels like a section that should appear in a long-form article on ESM or MSA Transformer. Are these results good compared to baseline methods? How does a PSSM perform on them? Is the goal to show that you are outperforming a baseline, or are you just doing some sanity checks for the model?


*****Updated review after discussion period*****
I have decided to raise my review to a weak accept, after a lengthy discussion period with the authors and between the reviewers. Overall, there are a number of detailed choices that were important to achieve the paper's results, and the community should know about these.

My principal hesitation is not around the merits of the paper's contribution, which is valuable, but in how the contribution is framed. If the paper is accepted, I would really appreciate it if you considerably changed the exposition (including the title) to be more precise about what's new in the paper. Specifically, I'd refer to your work and prior work (e.g., DeepSequence) both as zero-shot.

The title 'Language models enable zero-shot prediction of the effects of mutations on protein function' suggests that there is a new capability that they have 'enabled.' This is certainly not the case. Prior work uses exactly the same data (naturally-occurring sequences, but no experimentally-labeled data) and achieves comparable results. When I first started working on ML for proteins, I was enthralled when I learned that both protein structure and the functional effects of mutations could be predicted by inspecting a statistical model of homologous sequences. This is a remarkable result, and I'm concerned that the twitter-verse will mistakenly think that this particular paper unlocked it (it's actually been known for many years).




**Time Spent Reviewing:**

3

---

> ### Author Response · Authors · 2021-08-11
> **Response to Reviewer bbgP**
>
> Thank you for the comments, suggestions, and detailed review. We’re glad you agree that this is an important line of work and of interest to the community. The main critique is around (a) novelty; (b) significance; and (c) the definition of zero-shot prediction.
>
> *Significance*
>
> > This paper is part of a really important line of work on deep learning + proteins. My hesitation is around whether this paper should appear as a stand-alone paper at NeurIPS, not whether the results are of interest to the community.
>
> Previous work on unsupervised mutation prediction has been published at top venues, including Nature Methods (DeepSequence) and Nature Biotechnology (EVMutation). Each of these works applies an existing model (VAE or Potts Model) to the unsupervised mutation prediction task. These methods are widely used and were presented as stand-alone papers.
> In our work, we show state-of-the-art results in this problem setting while introducing a new pre-trained model (ESM-1v) and a new evaluation setting (zero-shot prediction). We find that new innovations are necessary for protein language models to perform well on this problem. We also perform extensive evaluation and ablations to support the development of the final model.
>
> *Zero-shot prediction*
>
> > I disagree with your definition of zero-shot in the paragraph starting on line 75.
>
> We use the definition as it is commonly used in the NLP community, see e.g. Brown et al. 2020. Table 3 in the Appendix summarizes the formal settings and representative methods. Current SOTA methods such as DeepSequence and EVMutation are trained on a protein by protein basis. The major difference in our work is that we show a single protein language model can be pretrained on millions of protein sequences across many diverse protein families and then be directly applied to predict mutational effects without further supervision.
>
> > Why is a method that relies on querying a sequence database to retrieve homologous sequences qualitatively different than your approach?
>
> Please consider Figure 2 which highlights the difference between the steps for our method (just model inference), and classical unsupervised (per family: sequence search to construct MSA, train a model from scratch for many GPU-hours).
>
> > Doesn't the training set for ESM-1v or MSATransformer include the same 'weak supervision' that the traditional approach relies on?
>
> In the traditional approach the model is given information on what are the relevant sequences for the specific prediction task in the form of the MSA. By contrast protein language models do not get this information through pre-training.
>
> To make these points more clear, we will update the discussion of zero-shot and few-shot transfer in Section 2.
>
> *Novelty*
>
> > My primary negative critique of this paper is its lack of novelty.
> > It has been previously established in the literature by many papers that the functional effect of mutations can be predicted in a zero-shot setting.
>
> We strongly disagree and address this criticism in the response to all reviewers. No prior work has systematically studied how protein language models can be used for unsupervised mutational effect prediction or developed methods to apply them to this problem. Prior state-of-the-art methods DeepSequence and EVMutation must be trained on a protein by protein basis and do not use pretraining at all.
>
> > One argument for a paradigm shift due to your approach is that practitioners don't have to train a new model for a protein family of interest…
> > However, it’s rare for practitioners to be considering a large number of families at once.
>
> Inference across multiple families is an important problem. Proteome scale inference is important and of interest, see for example recent work Frazer et al. 2020 on clinical interpretation of genetic variants. Furthermore attempts at large scale computational design have been made (Rocklin et al. 2017) in which 10,000 point mutants were generated from thousands of parent proteins. We believe our approach will further enable these types of large scale assays as well as accelerate the more typical workflow of single family design.
>
> *Additional questions*
>
> > Scoring schemes
>
> We evaluated four scoring schemes, including ones in which the likelihood changes at non-mutated positions are considered. We found that P(x_t | x_{-t}) performs best. In this scheme, probabilities are extracted according to the mask noise during pre-training. Please see Appendix A for the experiments and additional motivation. We will revise the paper to make this more clear.
>
> >  Epistasis
>
> We consider a variety of scoring schemes including pseudolikelihood which can incorporate epistasis.
>
> > To me, changing the training data does not amount to introducing a new model with the degree of originality characteristic of a stand-alone paper.
>
> Our work highlights the importance of the pretraining data distribution, and establishes that this is in fact of critical importance in the design of protein language models. We find a significant effect for clustering at different levels of sequence similarity on performance.
>
> > It's not clear what an application setting would be where practitioners need to make predictions for many protein families at high throughput.
> > The goal of this zero-shot prediction is to avoid needing to run wet-lab experiments
>
> We disagree. Prediction of variant effects has applications beyond protein design of single families. See for example Frazer et al. 2020 on clinical interpretation of genetic variants. See Rocklin et al. 2017 for protein design at scale.
>
> > Evaluation Metrics.  Can you please explain why spearman correlation is a reasonable performance metric?
>
> Spearman correlation is the primary performance metric used by prior work in this area, including EVMutation (Hopf, et al. 2017); DeepSequence (Risselman, et al. 2018) and Hsu, et al. 2021. Spearman correlation measures the correlation between the rank variables of two populations. As we do not supervise with experimental measurements, we cannot expect the predictions from the model to exist in the same range as the experimental measurements. Therefore, we focus instead on the relative ranking. This is also the more practical metric from a protein engineering perspective, as we are usually interested in identifying mutations with the highest or lower fitness.

---

> > ### Comment · Reviewer_bbgP · 2021-08-17
> > **thanks for the detailed response**
> >
> > As I'm reading the authors' response and also the other reviewers, it strikes me that there may not be consensus around what exactly are the SOTA performance results of the paper. Can you please confirm that I am interpreting things correctly?
> >
> > 1) ESM-1v in zero-shot mode matches (but does not exceed) the performance of DeepSequence on 41 DMS datasets.
> > 2) The zero-shot advantages of ESM-1v vs. DeepSequence are purely computational: both use the same (unlabeled) data, but DeepSequence does not need to be trained on a per-protein basis. ESM-1v avoids this by pretraining across a large number of proteins. I'm still not convinced that this is a huge win. Tools like jackhmmer can be run in parallel at scale quickly and return a model (a profile HMM). No further training is necessary.
> > 3) ESM-1v outperforms other recently-published deep protein language models when used in the zero-shot setting.
> >
> >
> > Fyi, It's possible that your analysis of (1) is restricted by the use of spearman rho. I understand that you do this because prior papers have, but I don't think it is a good metric, and I think your research would be improved by moving beyond it. For example, in certain settings it would be useful to detect mutations that of certain classes: deleterious mutations, fitness-improving mutations, mutations with neutral effect. When ranking by the model's scores, can you detect these classes? How about AUC, precision@k, etc?
> >
> > Further, when I asked about epistasis I wasn't asking about the model's capability to capture epistasis, but whether the model actually captures it in practice on your data. You can take the evaluation data and find epistastic multi-mutant variants that are non-functional but composed of mutations that led to functional variants when used individually. Is your model able to detect this? Doing so is a big selling point for Potts models vs. profile HMMs, for example. Are you able to match this?

---

> > > ### Author Response · Authors · 2021-09-01
> > > **Re**
> > >
> > > 1. Agree. We note it is a new result that models that are not specialized to protein families, can perform on par with SOTA specialized models.
> > >
> > > 2. Agree there are computational advantages. On PSSMs: they can be obtained relatively rapidly; however we show in Table 1 that the language models outperform them. The differences go beyond computational. For instance language models have different scaling properties than MSA based models. They are also multi-task, in that a single model can be used for structure, function, homology detection etc. Our results have implications generally for development of large scale language models as multi-task models for proteins.
> > >
> > > 3. Agree.
> > >
> > > On the ideas around epistasis and metrics: agree these are interesting topics; they are however open ended and beyond the scope of the current paper. We agree they are promising avenues for future work.

---

### Official Review · Reviewer_FVbp · 2021-07-08

**Rating:** 5
**Confidence:** 3

**Summary:**

Unsurprising results and the method is not novel.

**Limitations And Societal Impact:**


The language models are not proposed by this paper. It is not clear what is the contributions of this paper except for the conclusion that language models can partially predict the protein function.

We also expect to see if language models can provide some interesting biological insights.

**Main Review:**

Modeling the impact of amino acid mutations on protein-protein interaction plays a crucial role in protein engineering and drug design. But the structures of different families of proteins vary a lot, hindering the generalization of a machine learning model. This paper found that, without any supervision from experimental data or additional training, protein language models capture the functional effects of sequence variation, achieving state-of-the-art zero-shot prediction performance.

However, it is somewhat expected. In principle, language models learn the distribution of the protein sequences that are stable. The unstable sequences are usually recognized by language models as low probabilities of existence.

The introduction of the used prediction models is missing. It is hard to understand the acronyms without explanation.

The implementation details are not clear. What are the hyperparameters and how do you finetune them?

Have the authors compared the deep learning models with the traditional evolutionary conservation profiles?


**Time Spent Reviewing:**

2

---

> ### Author Response · Authors · 2021-08-11
> **Response to Reviewer FVbp**
>
> >However, it is somewhat expected. In principle, language models...
>
> Thank you for your comments. It is a reasonable hypothesis that language model probabilities will correlate with functional effects of sequence variation --  that’s why we performed this study! However, no prior work has shown this or systematically studied this for protein language models. In addition, we show that multiple innovations are required to achieve state-of-the-art zero-shot prediction, including a different sequence identity threshold for training data.
>
> > The introduction of the used prediction models is missing...
>
> Thank you for the suggestion. To introduce the prediction models more systematically, we will incorporate a section introducing each model in the revision.
>
> > The implementation details are not clear. What are the hyperparameters and how do you finetune them?
>
> A number of implementation details (including different methods for scoring of mutants, model selection, treatment of synonymous mutations, etc.) are present in the supplement (Appendices A and D). Different hyperparameters for these are tested on the validation subset (see Section 4.1). The method with highest correlation was then tested on the remaining mutational scanning datasets. If there are specific implementation details you would prefer to see moved to the main paper, please let us know.
>
> > Have the authors compared the deep learning models with the traditional evolutionary conservation profiles?
>
> Yes, we compare against both traditional and deep learning baselines in Table 1. Our baselines include evolutionary conservation profiles (PSSM), Potts models (EVMutation), and VAE methods (DeepSequence). For both the Potts model and VAE methods we additionally replicate the original training using MSAs generated from more recent sequence databases in order to maximize performance of the baselines.

---

> > ### Comment · Reviewer_FVbp · 2021-08-18
> > **Thanks for your response**
> >
> > I have read the author's response and the other reviews. But I am still confused about the key contributions of this paper.
> >
> > First, this paper claimed that " this work is the first to consider the unsupervised mutational effect prediction problem systematically with protein language models". However, Riesselman et al. have studied this problem in 2019.
> >
> > Second, this paper claimed that "this work is the first to develop a protein language model that equals the performance of SOTA mutational effect prediction methods", but this paper did not propose any novel protein language models.
> >
> > Considering the above concerns, I insist on my judgment.
> >
> >
> > Riesselman, A., Shin, J. E., Kollasch, A., McMahon, C., Simon, E., Sander, C., ... & Marks, D. (2019). Accelerating protein design using autoregressive generative models. BioRxiv, 757252.

---

> > > ### Author Response · Authors · 2021-09-01
> > > **Re**
> > >
> > > 1. Riesselman et al. 2019  is fundamentally different from our work. Riesselman et al. trains a specialized model for every protein family. In this paper we show that a general purpose model (i.e. not needing to be specialized for every task) can be trained once and then applied across many mutational effect prediction tasks. This is a significant advance and novelty in relation to prior work including Riesselman et al.
> > >
> > > 2. ESM-1v is a new model. We compare to prior models, e.g. ESM-1b in Table 2. As can be seen in Table 2, the performance of ESM-1v is higher than previous protein langage models.

---

### Official Review · Reviewer_85hq · 2021-07-16

**Rating:** 6
**Confidence:** 3

**Summary:**

The paper analyses the generalization capacity of the protein language models. The experiments and analysis suggest that zero-shot inference of protein language models can capture the functional effects of sequence variation. The paper is essentially analyzing the protein language models on their generalization capacity of zero-shot transfer learning for the Mutation Effect Prediction task.



**Limitations And Societal Impact:**

The deep neural network-based language models are good at predictions. However, they are prone to adversarial attacks, meaning that changing input slightly can change the predictions. Inference based on such a model can be problematic for a sensitive task like designing protein in the current state of the models.

**Main Review:**

The paper analyses the recent advancements of protein language models (PLM) on a specific task to check the generalization capacity of a PLM. However, the paper does not discuss the robustness of the zero-shot predictions. An error analysis will be really useful for understanding the true generalization capacity of the models.

There is no methodological novelty in the approach. It was also difficult to read the paper as a lot of reference to Appendix and supplementary materials.


**Time Spent Reviewing:**

Four hours

---

> ### Author Response · Authors · 2021-08-11
> **Response to Reviewer 85hq**
>
> > Robustness and error analysis
>
> Thank you for the notes on robustness and error analysis. We attempt to answer the robustness and generalization capability of the models in several ways. In Figure 3, reported results are the mean + standard deviation of spearman rho over 20 bootstrapped samples per DMS dataset, and error bars are reported. In addition to this error reporting, Section 5 explores the model’s predictions from a biological perspective to ensure that they are reasonable. We find that model predictions reflect conservation and hydrophobicity, and probability predictions of each amino acid are well calibrated.
>
> > Adversarial Attacks
>
> Given that our model is trained with a masked language modeling objective, which incorporates random corruptions of an input, small changes to the input are likely to be within the modeling distribution. In Appendix A, we evaluate a “mutant marginal” scoring method which involves inference over sequences with the mutations directly applied. We find that probabilities extracting from the model under this scoring method correlate nearly as well with the groundtruth data as our best performing scoring method (|Spearman rho| of 0.578 vs. 0.582 on the validation set).
>
> > There is no methodological novelty in the approach.
>
> We strongly disagree and discuss in detail in the comment to all reviewers. Methodological novelties of the paper include the introduction of a new protein language model, ESM-1v, that performs at SOTA on mutation prediction in its zero-shot mode, a number of scoring methods for protein language models (Appendix A); showing how to effectively unsupervised fine-tune ESM-1v (Appendix B); examining the effect of pre-training data and ensembling on the results (Appendix C); highlighting the biology learned by the models (multiple central figures); and achieving SOTA results on the MSA Transformer by examining a number of MSA subsampling strategies.
>
> > It was also difficult to read the paper as a lot of reference to Appendix and supplementary materials.
>
> We will make the link between the Appendix and paper more clear in the revised version of the paper.

---

> > ### Comment · Reviewer_85hq · 2021-09-01
> > **Re: Response to Reviewer 85hq**
> >
> > Thanks for the responses. I am still finding difficulty understanding the novelty of the proposed method. Moreover, the results and robustness analysis based on the correlation measurement --- 'Spearman rho' --- is a bit unconventional (if not inappropriate) given that the language modelling and predictive modelling have standard perplexity and accuracy based evaluation measurement. From the data mining perspective, related evaluation metrics could have been used.
> >
> >
> > I am inclining to keep my score unchanged.

---

> > > ### Comment · Reviewer_bbgP · 2021-09-01
> > > **choice of evaluation metrics**
> > >
> > > The goal of this paper is not to evaluate language models in terms of metrics defined on unsupervised data (e.g., perplexity), but to show that the language model is predictive of labels derived from supervised data (experimental measurements of protein fitness). This 'zero-shot' task has been around for a while in the protein community. This sort of prediction is very helpful for applications, since meaningful predictions about the functional effects of mutations can be made without performing expensive wet-lab experiments. It's also useful for methods development because it provides a way to benchmark language models on a relevant downstream task.

---

> > > > ### Comment · Reviewer_85hq · 2021-09-01
> > > > **Re: choice of evaluation metrics**
> > > >
> > > > Yes, I agree that the goal of the paper is not to evaluate language models. However, as the authors' mention (quoted below) in rebuttal that one of the contributions is the introduction of a new protein language model, I was expecting some evaluation measurement on the quality of that language model. In a domain related paper, titled "MSA Transformer", the perplexity was reported.
> > > >
> > > > Re: "Methodological novelties of the paper include the introduction of a new protein language model, ESM-1v, that performs at SOTA on mutation prediction in its zero-shot mode, a number of scoring methods for protein language models (Appendix A)"

---

> > > > > ### Author Response · Authors · 2021-09-01
> > > > > **Perplexities**
> > > > >
> > > > > Please see supplemental table 12 which presents perplexities on held out sequences. We will add a brief discussion of these results to the main paper.

---

### Official Review · Reviewer_kUmC · 2021-07-16

**Rating:** 7
**Confidence:** 3

**Summary:**

The paper describes a model that
- Is pretrained jointly across many protein families
- Learns functional effects of mutations at state-of-the-art for deep mutational scans
- Recapitulates substitution patterns concordant with biology
- Does not require MSAs at all

**Limitations And Societal Impact:**

The authors have sufficiently described potential negative societal impact.

Re: limitations: Questions for author:
- Why is it that fine tuning ESM-1v didn’t statistically improve performance across the 41 DMS tasks via t test?
  - What sort of level of improvement would be needed for this?
  - Are there statistically significant claims you can make about the fine-tuning performance?
  - Why was there such a big jump from fine tuning for some datasets but not others?
  - There are still some tasks where fine-tuned performance is statistically lower than DeepSequence. Do you have any explanations for this?
- Is there a baseline you can compare to for calibration error? It’s a little difficult to understand whether ACE of 0.006 is good. How difficult is it to create a model that does this well?
- What is the p value on pearsonr for PSSM entropy and predicted entropy?
- Saying pearsonr of .44 on the entropy claim is well correlated is slightly hyperbolic - I suggest wording.
- Is the main claim that this is faster than current methods? Or faster to train? Fewer parameters? If so, a comparison to DeepSequence in this would be helpful.
  - Perhaps this is already in the literature (transitively?) and I don’t know about it, in which case a reminder in the text would be useful.
- A connection between calibration and why it’s important for downstream tasks would help the paper’s sales pitch.
- Figure 6: Average time to compute a DMS is a slightly odd statistic, as this measure is quite dependent on somewhat extraneous factors like the assays used in the experiments. I suggest normalizing a different way, e.g. per sequence or per residue.


**Main Review:**

This paper solves the problem of functional mutation effect prediction, and does it as well as anyone else, with a very simple setup. They demonstrate these results convincingly. My main concern with this paper is that they achieve these results in a zero-shot manner, but don’t demonstrate that fine-tuning improves performance, which is very surprising. That said, I do not think that they need to completely answer the question as to why in order for this to be a complete body of work.

Originality: Not very original methods. Use of current dataset.
Quality: Good quality. Lots of qualitative analysis. Lots of baselines. Baselines were tuned beyond my expectation.
Clarity: clear. Good prose.
Significance: Zero-shot performance mirroring state-of-the-art in an area that has substantial impact, like pharmaceuticals, is very significant.

See below for more specific details.

-----------------------------------------------

The paper describes a model that
- Is pretrained jointly across many protein families
- Learns functional effects of mutations at state-of-the-art for deep mutational scans
- Recapitulates substitution patterns concordant with biology
- Does not require MSAs at all

The evidence they provide is
- _Is pretrained jointly across many protein families_
  - Use UniRefXX for many values of XX
- _Learns functional effects of mutations_
  - Quantitative evidence via deep mutational scans (DMS)
    - 41 DMS examined
    - Spearman R used - SOTA values
    - Comparison to DeepSequence, ESM-1b, ProtBERT-BFD, UniRep, TAPE
  - Show entropy vs ground truth PSSMs
  - Show calibration of the model
- _Recapitulates substitution patterns concordant with biology_
  - 41 DMS examined
  - Explanations of substitution patterns like hydrophobicity and buried vs exposed via some anecdotes and statistics
- _Does not require MSAs at all_
  - Via an explanation of their architecture, training procedure, and inference setup

The benefits of their approach vs some SOTA are
- No fine tuning step needed
- No family-based training objective function
- Not trained for-purpose - should be generally useful.


**Time Spent Reviewing:**

4

---

> ### Author Response · Authors · 2021-08-11
> **Response to Reviewer kUmC**
>
> Thank you very much for your review and your detailed comments. We will incorporate the notes suggested and have attempted to answer the questions below.
>
> > Originality: Not very original methods. Use of current dataset
>
> This paper is the first to systematically study unsupervised mutational effect prediction with protein language models. We show that prior SOTA protein language models e.g. ESM-1b, ProtBERT-BFD, TAPE, Unirep, do not perform at SOTA on this problem and perform extensive experiments to understand how the models can be improved to achieve SOTA performance. We discuss the novelty of the paper in more detail in the response to all reviewers.
>
> > Why is it that fine tuning ESM-1v didn’t statistically improve performance across the 41 DMS tasks via t test?
>
> The fine-tuning in these experiments is *unsupervised* i.e. on sequences only. ESM-1v is pretrained across millions of diverse protein sequences. During the fine-tuning procedure, we only use sequences from the MSA, no experimental measurements of function are used. One can think of the fine-tuning procedure as focusing the model’s distribution toward the protein of interest. Our findings that fine-tuning did not statistically improve performance across the 41 DMS tasks suggests that the model has already captured the necessary information during the pre-training stage. While previous work in this area requires models to be trained on MSAs, we find that ESM-1v works out of the box in a zero-shot manner. Fine-tuning is not required.
> There are a number of possible ways to perform unsupervised fine-tuning. We explored these extensively, finding little benefit despite trying a number of different approaches. These experiments are detailed in Appendix B.
>
> > Is there a baseline you can compare to for calibration error? It’s a little difficult to understand whether ACE of 0.006 is good. How difficult is it to create a model that does this well?
>
> We are not aware of any protein language modeling papers which report calibration errors. More broadly, we are not aware of many deep learning papers that report calibration errors at all. Multi-class calibration metrics appear to be very rarely reported at all and even the metric we report comes from two relatively recent papers. We would welcome any proposed baselines by the reviewers.
> To give more intuition on what the calibration errors mean, first note that as a sanity check we are able to detect the model’s bias towards Methionine (start codon) in the first position. Virtually all sequences used during pre-training begin with Methionine, so this is expected. Second, it appears there is not a similar bias towards other amino acids or at other positions. Finally, we hope that reporting this metric here will at least serve as a baseline for future methods and comparisons.
>
> > What is the p value on pearsonr for PSSM entropy and predicted entropy?
>
> Note the huge sample size (15k sequences x average length of 300 = 4.5M amino acids).
>
> > Is the main claim that this is faster than current methods? Or faster to train? Fewer parameters? If so, a comparison to DeepSequence in this would be helpful.
>
> Zero-shot transfer using protein language models represents a new approach to unsupervised mutational effect prediction. This has practical benefits in terms of inference speed, and settings in which the models can be used. I.e. it is fundamentally different that training a new model on a protein by protein basis which is the current standard practice for unsupervised mutational effect prediction. It also has implications for our understanding of protein language models -- e.g. it indicates that information about protein function is captured by these models even though they have only been trained on sequences.

---

### Official Review · Reviewer_Bsx3 · 2021-07-18

**Rating:** 8
**Confidence:** 4

**Summary:**

This paper demonstrates utilizing a pretrained masked language model to predict a ranking of functional activity of a protein.

They focus on data from deep mutational scanning, which measures the functionality of a large set of individual mutations to a protein in order to characterize the fitness landscape around the protein (wild type) for the desired functionality.

It proposes a method which uses the log odds ratio between a mutated variant and the wild type protein as a proxy for the fitness of the mutation. It evaluates the methods by using Spearman correlation between its ranking and ground truth fitness from deep mutational scanning datasets.

It evaluates a wide set of existing baselines, and focuses analysis on an updated version of the ESM-1b model, ESM-1v. It compares several methods for evaluating the log odds of a particular mutation.


**Limitations And Societal Impact:**

The paper's appendix discussion focuses on compute costs.  While far off, it is worth discussing potential applications of this technology (good and bad), even if just referencing existing writing on potential dangers of protein engineering.

**Main Review:**


I am primarily evaluating the method from the perspective of a machine learning researcher who has done related work applying self-supervised learning methods to protein data.

Overall, the method is well motivated, simple, effective, and likely to lead to fruitful future research. The method is carefully evaluated against several baselines.

**Originality**: The datasets are standard benchmarks for evaluating protein function prediction. The models are variants on standard BERT models, with evaluations to select the right pretraining dataset and compare model sizes.

It demonstrates a simple new method on top of this established model to accomplish a core task in protein design (characterizing function of mutations to a protein) with minimal overhead.

It performs about as well as methods that use a full MSA while only requiring a single sequence.

**Quality**: The paper is technically sound and carefully evaluated. It answered questions I had while reading it with experiments (and a thorough appendix).



* Proper validation sets are used
* Baselines are well tuned (even outperforming results in source publication)
* results are reported using sound statistics
* The ESM-1v model is evaluated at several different scales and across many pre-training datasets
* Several alternatives are considered for the primary choices made in the paper (e.g. how to score the effects of mutations given the model, which pre-training dataset to use, how to
* Good discussion of related work, in both ML + protein and NLP literature.
* Extends the model analysis beyond the task at hand to how the models do at modeling the stability of residues located in different regions of the folded protein.

**Clarity:** The paper is clearly written, with sufficient references to related work and background. Figures clarify content of the text (such as the overall task, the pipeline for each baseline and the proposed method, and visualizations of relationship between position and model entropy).

I believe a practitioner familiar with the data and BERT models could reproduce the results as described.

**Significance:** I believe other researchers are likely to build on the methods outlined in the paper. While it does not always exceed the baselines, the method generally performs near or better than existing methods despite only needing a single protein sequence (instead of a full MSA). It is also the first paper I have seen that explicitly evaluates scaling on protein datasets (which is why I would like to see more discussion of this point).

**Feedback for improvement**

**– content –**



* Consider saying "masked language models" in the title to point at the specific model class used
* line 210: What methods are appropriate for handing Methionine in the first position ("care must be used when applying the model to a subsequence")
* Model
    * Any theories as to why uniref90 performs the best?
    * Does the pretraining only use the cluster centers from each cluster? Consider adding a section to appendix expanding on the pretraining details, even if it is explained in a referenced paper
    * How large was the dataset? Roughly how many epochs was 170k steps of 1 million tokens
    * figure 9: Add  figure or table including other pretraining metrics alongside just the plot of rho with respect to model size/update steps. What is the per amino acid accuracy? Perplexity? table 12 has some of this for the large model, but the other model information is also useful.
        * Generally suggest trying to characterize relationship between pretraining performance and downstream performance on your task as measured by rho.
* I'm sure I've seen similar methods to "spiked unsupervised fine-tuning" used in NLP literature to regularize/combat forgetting, but I'm struggling to find a reference… I will see if I can find something for the discussion period.
* Equation at line 92:
    * Define mt=mutated, wt=wild type before equation
    * should the conditioning be x^{mt}_{-t} and x^{wt}_{-t}?
* 96/104: consider describing more about what fitness landscape means (fitness of the region around the protein in sequence space).
* 526: The #s don't add up:"among the 15% predicted positions, 10% are randomly, 10% retain …"

**– style –**



* 54: add comma after "in biology"
* Define MSA when it is first used (79)
* Line 94: You do not describe what ESM-1v before referring to it
* table 7: misspelled marginal
* figure 9
    * label the y axis
    * The bottom of the figure is cut off
* table 1: Add reference to appendix section where you describe sampling for the MSA

**Time Spent Reviewing:**

4

---

> ### Author Response · Authors · 2021-08-11
> **Response to Reviewer Bsx3**
>
> Thank you very much for your detailed review and feedback.
>
> > Any theories as to why uniref90 performs the best?
>
> Variant effect prediction relies on accurately predicting the effects of relatively small changes to an existing protein. As a result, pre-training datasets that include more densely sampled sequences (such as UniRef90 and Uniref100) could perform better. The higher density of sampling should help the model learn which mutations are more likely in a smaller neighborhood around a protein. It is unclear why UniRef90 is better than UniRef100. This finding seems to imply that too much redundancy is harmful and that as in structure prediction reweighting the data distribution toward a more diverse set of sequences is important.
>
> > Does the pretraining only use the cluster centers from each cluster? Consider adding a section to appendix expanding on the pretraining details, even if it is explained in a referenced paper
>
> Pretraining uses the cluster centers from each cluster. Thank you for the suggestion. We will add a section to the appendix expanding on the pretraining details.
>
> > How large was the dataset? Roughly how many epochs was 170k steps of 1 million tokens
>
> Uniref90 consists of 113 million diverse protein sequences, total of 38.7 billion tokens.  After a 10% heldout split, this means about 35k steps = 35 billion tokens per epoch. We will update the paper with this information.
>
> > figure 9: Add figure or table including other pretraining metrics alongside just the plot of rho with respect to model size/update steps. What is the per amino acid accuracy? Perplexity? table 12 has some of this for the large model, but the other model information is also useful.
>
> Thank you for the suggestion. We will update the paper with additional pretraining metrics for the model scaling experiment. We will also update Table 12 to include the per amino acid accuracy.
>
> Thank you again for the suggestions to improve the clarity of the paper. We will consider and incorporate these suggestions into the paper revision.

---

> > ### Comment · Reviewer_Bsx3 · 2021-08-31
> > **How to handle M in first position?**
> >
> > Followup on one of the points in my review. Wondering about methodology here.
> >  line 210: What methods are appropriate for handing Methionine in the first position ("care must be used when applying the model to a subsequence")

---

> > > ### Author Response · Authors · 2021-09-01
> > > **Re. How to handle M in first position**
> > >
> > > It will depend on the application. We’ve chosen to highlight the bias toward M in the first position without being prescriptive about how to handle it. Possibilities could include appending methionine to sequences without it, excluding the first position, providing a token to distinguish the sequence start from a crop, removing the starting methionine in pretraining, etc.

---

### Author Response · Authors · 2021-08-11
**Response to all reviewers**

We thank the reviewers for thoughtful comments and suggestions. We note that many of the reviewers found this to be an interesting work and expect it to be impactful.

For example: “Overall, the method is well motivated, simple, effective, and likely to lead to fruitful future research.” (Bsx3). “This paper solves the problem of functional mutation effect prediction, and does it as well as anyone else, with a very simple setup. They demonstrate these results convincingly…  Zero-shot performance mirroring state-of-the-art in an area that has substantial impact, like pharmaceuticals, is very significant.” (kUmC). “The experiments and analysis suggest that zero-shot inference of protein language models can capture the functional effects of sequence variation… The deep neural network-based language models are good at predictions” (85hq). “But the structures of different families of proteins vary a lot… this paper found that, without any supervision from experimental data or additional training, protein language models capture the functional effects of sequence variation, achieving SOTA zero-shot prediction performance” (FVbp). “The new task (zero-shot scoring of the effects of mutations) has a variety of useful applications for clinical medicine and protein design… A paradigm shift due to your approach is that practitioners don’t have to train a new model for a protein family… This paper is part of a really important line of work on deep learning + proteins.” (bbgP).

A few reviewers raise the novelty of the paper as a concern. We strongly disagree with this critique and address it directly here. Prior work on protein language models has not considered the unsupervised function prediction problem studied in this paper. Some prior work has looked at *supervised* prediction of mutational effects, e.g. Alley et al. 2019, Rives et al. 2019, Rao et al. 2019, Hsu et al. 2021. However work on large scale protein language models has not considered the significantly harder *unsupervised* prediction problem. These are fundamentally different problem settings. In the supervised setting models are trained with experimental measurements of protein function. Such data is not available for the vast majority of proteins, and makes unsupervised prediction methods important. Some of the points of novelty in this work include:

1. This paper is the first to consider the unsupervised mutational effect prediction problem systematically with protein language models. It establishes that prior SOTA protein language models, e.g. ESM-1b, ProtBERT-BFD, TAPE, Unirep, fall short of the current SOTA mutational effect prediction methods. No other prior work has systematically looked at unsupervised mutational effect prediction with protein language models.

2. This paper is the first to develop a protein language model that equals the performance of SOTA mutational effect prediction methods. This implies that these models can be used by practitioners.

3. This paper demonstrates that SOTA performance can be obtained with zero-shot and few-shot transfer, i.e. without any specialization to particular proteins. This is a major point of departure from prior work, e.g. DeepSequence and EVMutation, the current SOTA methods, which must be trained on a protein by protein basis. By contrast the models developed here are trained once and then can be applied directly to any protein. This is novel from a methodological standpoint, and has practical benefits in settings where inference at scale is useful, for example in large scale functional assays (Rocklin et al. 2017) or proteome scale clinical variant prediction (Frazer et al. 2020).

---

### Decision · Program_Chairs · 2021-09-27

**Decision:**

Accept (Poster)

**Comment:**

While evaluating this paper, the reviewers had an extensive discussion about the relative strengths, and in particular what counts as "novelty".  While the reviewers did not come to unanimous consensus here, I am swayed by the majority of reviewers who cited the strong empirical rigor in analyzing the behavior of the proposed models, in addition to the careful combination of many small improvements that lead to strong results on an important application area in predicting the functional effects of protein mutations.  Indeed, my own view is that the term "novelty" is often overly constrained to mean "a brand new modeling technique" when instead it should be interpreted to mean "something important that adds to the field's overall understanding and knowledge".  Indeed, the recent ML literature is filled with important papers that usefully shine new light on previously known modeling techniques or methods.

Given the strong reviews from the majority of reviewers, and the acknowledgement of the strong empirical results and careful explication of the rigorous evaluations, I am happy to recommend acceptance of this paper.  I do expect that the authors will use the extensive reviewer feedback to further revise and strengthen the paper for its final form.